# Fingerprinting the Cretaceous-Paleogene boundary impact with Zn isotopes

Ryan Mathur [1✉], Brandon Mahan[2], Marissa Spencer [3], Linda Godfrey[4], Neil Landman[5], Matthew Garb[6], D. Graham Pearson [7], Sheng-Ao Liu[8] & Francisca E. Oboh-Ikuenobe[3]

Numerous geochemical anomalies exist at the K-Pg boundary that indicate the addition of extraterrestrial materials; however, none fingerprint volatilization, a key process that occurs during large bolide impacts. Stable Zn isotopes are an exceptional indicator of volatility-related processes, where partial vaporization of Zn leaves the residuum enriched in its heavy isotopes. Here, we present Zn isotope data for sedimentary rock layers of the K-Pg boundary, which display heavier Zn isotope compositions and lower Zn concentrations relative to surrounding sedimentary rocks, the carbonate platform at the impact site, and most carbonaceous chondrites. Neither volcanic events nor secondary alteration during weathering and diagenesis can explain the Zn concentration and isotope signatures present. The systematically higher Zn isotope values within the boundary layer sediments provide an isotopic fingerprint of partially evaporated material within the K-Pg boundary layer, thus earmarking Zn volatilization during impact and subsequent ejecta transport associated with an impact at the K-Pg.

[1] Geology Department, Juniata College, Huntingdon, PA, USA. [2] Earth and Environmental Science, James Cook University, Townsville, QLD, Australia. [3] Department of Geosciences and Geological and Petroleum Engineering, Missouri University of Science and Technology, Columbia, MO, USA. [4] Department of Earth and Planetary Science, Rutgers University, New Brunswick, NJ, USA. [5] Division of Paleontology, American Museum of Natural History, New York, NY, USA. [6] Department of Earth and Environmental Sciences, City University of New York, Brooklyn College, Brooklyn, NY, USA. [7] Department of Earth and Atmospheric Sciences, University of Alberta, Edmonton, AB, Canada. [8] China University of Geosciences, Beijing, China. ✉email: mathurr@juniata.edu

Scientific debate regarding the global event that occurred at the Cretaceous–Paleogene (K–Pg) boundary, which ultimately rendered non-avian dinosaurs and many other lifeforms extinct, has persisted for nearly half a century. Multiple lines of geophysical, geochronological, and geochemical evidence identify the existence of a bolide impact and the geological processes associated with this collision. The identification of K–Pg layers containing anomalously high Ir (and other PGE) concentrations within K–Pg age sedimentary strata by Alvarez et al.[1] sparked large scientific efforts to discern clear geochemical signatures at the K–Pg boundary. Subsequently, this Ir/PGE anomaly has now been identified in well over 100 marine and continental sections worldwide[2–5], and the impact ejecta associated with the bolide impact (spherules, shocked minerals, Ni-rich spinels) have been well-characterized and documented across the globe[6].

Furthermore, an extraterrestrial impact component has been evidenced by the presence of a carbonaceous chondrite-like (CM2) Cr isotope anomaly ($\varepsilon^{54}Cr$)[7], and Os isotopes have indicated that the Deccan traps volcanism preceded the K–Pg boundary, thus largely negating volcanism as the main driver of the K–Pg boundary mass extinction event, though debate persists[5,8]. Geochronological data provide harmonious dates between Chixculub crater melt rock and K–Pg boundary ejecta (impact spherules and fragments)[9], and along with the presence of shocked mineral phases (e.g., quartz, feldspars)[10], large-scale geophysical studies pinpoint the Yucatan Peninsula as the paleo-geographical locus of the bolide impact[11,12]. Goderis et al.[13] provide a concise overview of the various geological/mineralogical, geochronological, geophysical, and geochemical lines of evidence in support of an impact event at the K–Pg boundary. Additionally, paleontological and paleoclimatic data support the notion that a bolide impact caused the abrupt climatic shift and mass extinction, with only a secondary to minor role for preceding and/or concomitant volcanism[14,15].

While there is an abundance of evidence supporting the impact hypothesis for the K–Pg boundary event, geochemical evidence in the literature provides indicators of a marked change in sediment and/or atmospheric composition across the K–Pg boundary, and thus this evidence is only circumstantially connected to the impact by temporal/stratigraphic association (as noted by the review of[6]). That is, they are phenomenological indicators (e.g., addition of anomalous/exotic materials), not mechanistic indicators (e.g., fingerprinting an impact-related process). In this way, the "impact hypothesis" for the K–Pg boundary fundamentally lacks a clear process tracer that can define bolide impact as the mechanism underlying such anomalies within the K–Pg boundary layer.

A number of transition metals are relatively volatile, and thus their application to tracing physico-chemical mechanisms associated with impacts—chiefly volatilization—may provide novel insights into these processes during bolide strikes and may provide a more definitive, relatively routine fingerprint of impact ejecta material. Of the stable transition metal isotope systems, Zn provides the most promise, with numerous published applications to impact sources and processes. Much of this work has been performed on tektites, where a wealth of literature has consistently shown that as heated projectiles (e.g., molten proto-tektites) travel along their ballistic trajectories, Zn volatilization results in lower Zn concentrations and heavy Zn isotope enrichment (high $\delta^{66}Zn$ in per mil, ‰) in the residuum[16,17]. Studies of nuclear blast ejecta[18] and super-heated impact melt sheets[19], as well as experimental work[20,21], all corroborate this generalization.

In the current work, Zn isotope compositions of K–Pg sedimentary rock layers have been characterized for five different locations, from proximal (up to 1000 km; Mississippi and Missouri sites) to intermediate distances from (1000–5000 km; Montana and New Jersey) the impact site at Chixculub crater in the Yucatan Peninsula. All stratigraphic sequences considered contain the characteristic spherule layer and ash associated with the impact[6,22] (Fig. 1). Samples from these sites originate from well-studied outcrops and drill cores, and the depositional environment for samples range from terrestrial (Montana), transitional (Missouri), and shallow-water marine (Mississippi and OPD Leg 165 in the Caribbean), to deeper water marine (New Jersey).

## Results and discussion

**Evidence for a distinct volatile Zn signature in boundary layer sediment.** The salient observation of the current dataset is that the boundary layer sediments possess high $\delta^{66}Zn$ values that correlate with low Zn concentration. The Zn isotope composition of the K–Pg layers analyzed is higher than that of the surrounding sedimentary rocks (an increase of +0.7 per mil at the Caribbean site, +0.3 per mil at the Missouri and Montana sites, +0.1 per mil at both the Mississippi and New Jersey sites (Table 1 and Figs. 1 and 2). Explanation of the origin of this signature requires a process that generates both characteristics. Multiple hypotheses could explain natural processes that generate higher Zn isotope values such as melt-induced fractionation, inherited signatures from the target rocks where the meteor struck, meteoritic materials that settled at the boundary, secondary alteration/depositional processes, and residuum from volatilized material from the impact event that settled in the "fall-out" layers. However, a process producing the combination of higher Zn isotope values with lower Zn concentration must occur to explain the trend observed here. The discussion below explores which of these processes might be capable of generating a high Zn isotope value and low Zn concentration end member, and/or could create a consistent Zn isotope signature over such large spatial scales.

Mechanisms relating to magmatic differentiation processes and associated with melting of target rocks at the impact site cannot generate the elevated Zn isotope values present in the boundary layer. Multiple studies of Earth's mantle melts, and their associated peridotite residues reveal that Zn isotopes do not routinely fractionate greater than 0.1‰[23–27] during magmatic processes at mantle potential temperatures of 1300–1500 °C. Considering that the instantaneous melting temperature associated with impact matches or exceeds this range (e.g., >1300 °C decelerative drag heating during terminal in-fall of ejecta[28]), the production of impact melt alone is unlikely to produce the relatively large Zn isotope fractionation measured herein (e.g., the Caribbean site samples).

Further evidence that magmatic processes did not cause the Zn isotope and concentration anomalies of the K–Pg reside in a comparison with the Permian-Triassic boundary (P–T). The P–T boundary possesses an abrupt increase of Zn concentration and a concomitant ~0.5‰ decrease in $\delta^{66}Zn$ occurring ~35 ky before the end-Permian mass extinction. The Zn isotopic and concentration changes have been interpreted to demonstrate rapid and massive input of isotopically light Zn from volcanic ashes, hydrothermal inputs, and/or extremely fast weathering of large igneous provinces (LIPs) into the oceans[29]. In comparison to the P–T boundary, the elevated Zn isotope ratios and decrease of Zn concentration of the sedimentary rocks in K–Pg boundary sediments are clearly inconsistent with such inputs of isotopically light Zn from processes associated with extensive volcanism. Under this premise, a mass flux from volcanism and weathering associated with the Deccan Traps, the LIP which precedes the K–Pg boundary[5], would only serve to dampen the positive $\delta^{66}Zn$

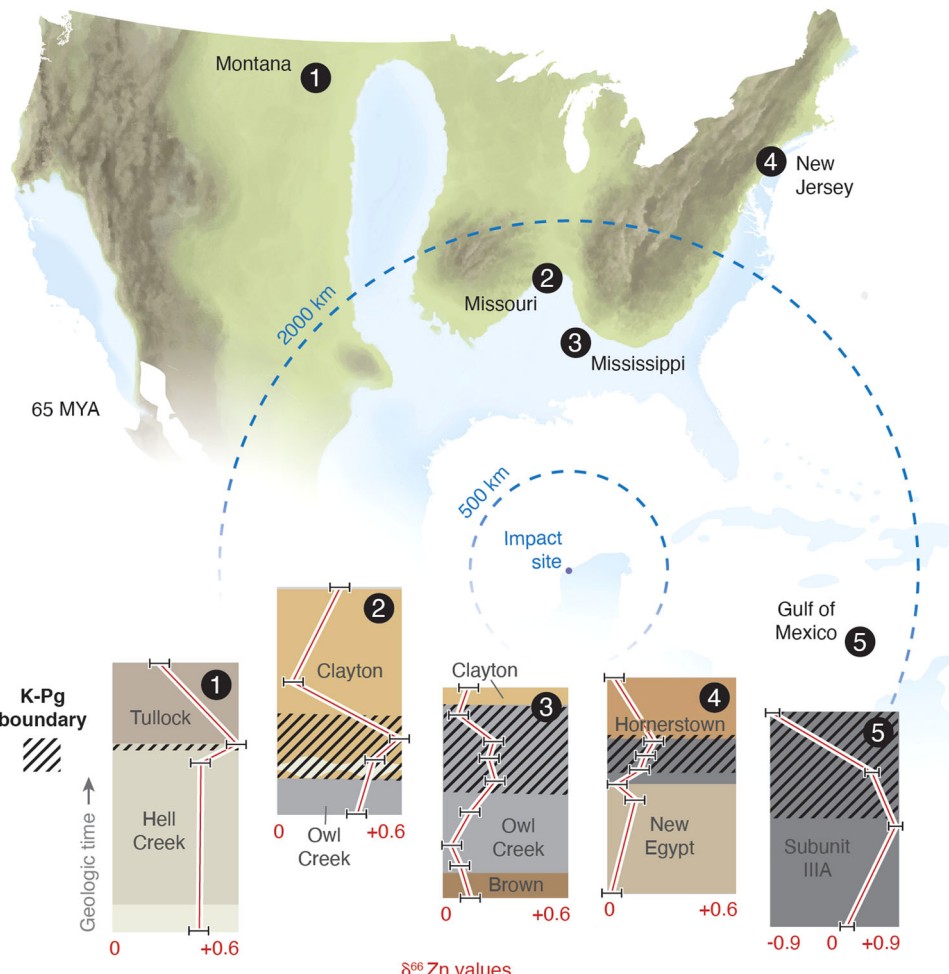

**Fig. 1 Paleo-depositional map of North America during the end of the Cretaceous.** Approximate locations of sampling sites plotted with the Zn isotope variations seen in the different stratigraphic intervals (formations labeled in cartoon stratigraphic column). Errors of the Zn isotopes are 0.05 per mil and are not significantly larger than the red line with white boarders. The cross-hatched area indicates the layers that contain the K-Pg boundary.

excursion at the boundary, not to generate or enhance it (especially in the marine settings).

Inheritance of the Zn isotope signature from target rocks does not explain the unique Zn signature in the K–Pg boundary layer. Extensive study of Zn isotope compositions in Earth materials over the last ~20 years indicates that the vast majority of terrestrial geologic processes do not significantly fractionate Zn isotopes beyond ~0.1‰, with the exception of volatility-related processes. While certain mineral, gas, and biologically derived materials may possess Zn isotope compositions outside the range reported above (e.g., $\delta^{66}$Zn values ranging from −1 to +2‰), bulk rock values containing such phases typically do not deviate from bulk silicate Earth or continental crustal values, because these phases are often only present at trace levels at the macro/regional scale. This is most relevant for biogenic carbonates that can have enrichments in heavy Zn isotopes[30], however these typically contain only ~1 ppm Zn[31], while isotopically lighter (lower $\delta^{66}$Zn) continental margin and basin marine sediments (e.g., $\delta^{66}$Zn ~0.12‰) can contain 10 to 100s of ppm Zn[32].

Inheritance of the Zn isotope signature from the meteorite impactor material(s) is also improbable as a source for the elevated Zn isotope signatures within the K–Pg boundary layer, because carbonaceous chondrites generally possess >100 ppm Zn, and typical Zn isotope values are less than +0.5‰[33] (with few outliers beyond this). The concentration data alone eliminate meteorites as a source, generally speaking, because the mixing

trends observed at the boundary layer in that case would have a trend opposite to what is observed herein. Notable exceptions in the literature, with respect to high $\delta^{66}$Zn values, are heated and/or thermally metamorphosed meteorites[16,34–37]. However, such examples—e.g., impact-heated or thermally metamorphosed meteorites[33,35]—are extremely rare in the meteorite record, and these meteorites are also generally marked by low Zn concentrations (as they are themselves likely products of Zn loss by volatilization). As a final point in this regard, Cr isotopes indicate a [CM2] carbonaceous chondrite-like material as the K–Pg bolide[33], thus excluding the latter possibility of a thermally metamorphosed [enstatite] chondrite as the high $\delta^{66}$Zn source, since enstatite (and ordinary) chondrites have dramatically different Cr isotope signatures relative to carbonaceous chondrites[38]. Given these observations, and the fact that impact ejecta is largely composed of target material (with higher Zn concentrations), it is exceedingly unlikely (though not entirely impossible) that such materials are the high $\delta^{66}$Zn source indicated herein.

Zn isotopic shifts related to secondary alteration processes are not significant enough to produce the Zn isotope excursions generally found at the K–Pg boundary. Redox and other kinetic reactions associated with biogeochemical interactions in soils and shallow depositional environments produce relatively small Zn isotope fractionations (0.1–0.2‰[32,39–41]), and measured Zn isotope fractionation among geochemical products and reactants

**Table 1 Zn isotope and concentration data from K–Pg sedimentary sites.**

| Sample | Location | $\delta^{66}Zn_{ETH}$ (per mil) | $\delta^{66}Zn_{Lyon}$ (per mil) | Zn (ppm) |
|---|---|---|---|---|
| Clayton Formation (lower), CI-2, ST10, MST-2185 | MI | 0.26 | 0.53 | 34 |
| Clayton Formation, Purina Plant, GL-G, Tr. 1, STC05 MST-2180 | MI | −0.13 | 0.14 | 557 |
| Clayton Formation, Purina Plant, Gl-L Tr. 1, ST06 | MI | 0.38 | 0.65 | 93 |
| Upper Clayton Formation, Purina plant, Tsunami deposit | MI | 0.10 | 0.37 | 159 |
| Owl Creek Formation, SCC25, TLE-5a, MST-2137 | MI | 0.18 | 0.45 | 63 |
| 135333 | MS | 0.02 | 0.29 | 116 |
| 135334 | MS | 0.05 | 0.32 | 106 |
| 135335 | MS | −0.16 | 0.11 | 156 |
| 135336 | MS | −0.1 | 0.17 | 152 |
| 135337 | MS | 0.06 | 0.33 | 173 |
| 135339 | MS | −0.2 | 0.07 | 121 |
| 135340 | MS | −0.15 | 0.12 | 166 |
| 135341 | MS | −0.09 | 0.18 | 118 |
| Spherule 1 | MS | 0.12 | 0.39 | 357 |
| Spherule 2 | MS | 0.16 | 0.43 | 384 |
| Bearpaw Formation, Snow Creek Site 1 | MT | 0.21 | 0.48 | 89 |
| Hell Creek Formation, Snow Creek | MT | 0.22 | 0.49 | 107 |
| K/Pg Boundary Clay, Seven Blackfoot | MT | 0.40 | 0.67 | 17 |
| 1259, New Egypt Frm | NJ | −0.01 | 0.26 | 252 |
| 1259.6, New Egypt Frm | NJ | −0.18 | 0.09 | 90 |
| 1260.07, Boundary Layer | NJ | −0.07 | 0.20 | 126 |
| 1260.2, Boundary Layer | NJ | −0.05 | 0.22 | 120 |
| 1260.4, Boundary Layer | NJ | −0.18 | 0.09 | 90 |
| 1260.55, Homerstown Frm | NJ | −0.10 | 0.17 | 344 |
| 1261, Homerstown Frm | NJ | −0.20 | 0.07 | 132 |
| 165 1001A 38R CC 13-15 CM, top of boundary | ODP | −0.83 | −0.56 | 13.03 |
| 165 1001A 38R CC 25-27 CM, middle of boundary | ODP | 0.83 | 1.10 | 164.5 |
| 165 1001B 18R 40-41 CM, bottom of boundary | ODP | 0.2 | 0.47 | 41.7 |
| 165 1001B 38R 12-13 CM, carbonate at base of boundary | ODP | 0.13 | 0.40 | 15.03 |
| 165 1001B 18R 5 30-32 CM, middle of boundary | ODP | 0.53 | 0.80 | 88.89 |

Location information has USA state abbreviations and ODP = Ocean Drilling Program site data.

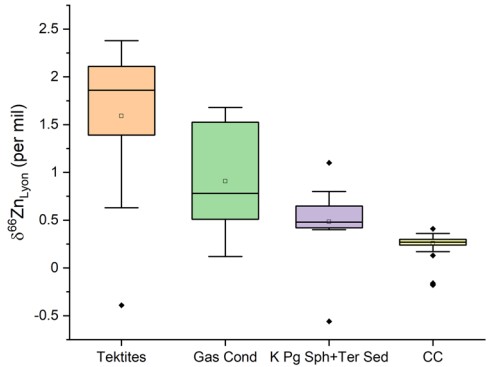

**Fig. 2 Box-and-whisker plot of the Zn isotope compositions of the different Zn isotope reservoirs.** Gas cond gas condensate measured from volcanic fumarole in Merapi[63], Sph spherules and Ter. Sedimentary rocks presented here. CC continental crust with the data taken from the literature[16,23,36,47,64]. Outliers are black symbols and lie outside the 95% confidence interval.

rarely approach or exceed 0.3‰ in ore minerals and clays[42–45]. The stratigraphic changes in Zn isotopic signature of the K–Pg boundary sediments (Fig. 1) also indicate minimal potential for terrestrial alteration as a cause for the higher $\delta^{66}Zn$ values at the boundary. Significantly, sediment layers above and below the interval sampled in the sections show no element-isotope mixing systematics. This indicates that no significant leaching of heavy Zn isotopes from under- or overlying rock layers took place, arguing against this as a cause of high $\delta^{66}Zn$ values within the

boundary layer itself. It also highlights the otherwise typical nature of the stratigraphic sequence with regards to Zn isotope compositions. These observations agree with studies that have addressed secondary fractionation processes in sediments, which indicate that such processes do not fractionate Zn isotopes in a systematic way or to any large degree[42,43,46]. As a final point here, it is unlikely that such local/regional scale processes would produce a consistent signal across the spatial scales and significantly different depositional environments considered herein.

Those things considered, the most probable explanation is that the high $\delta^{66}Zn$ values coupled with low Zn concentration observed at the K–Pg boundary originate from partial volatilization of Zn during impact-related processes (e.g., drag heating during ballistic outfall). Research on tektites—fused glasses from cooling of impact-induced melts[47]—along with complementary work on fused silicates from nuclear blast sites and experimental Zn volatilization show that Zn isotopes fractionate by as much as +0.5 to +2‰[19–21] during Zn evaporation. For impact ejecta, the physical composition of projectiles is often that of the target material mixed with a smaller contribution from the impactor, and therefore the initial Zn isotope compositions of projectiles most likely mimic that of the target material. During impact events, ejecta such as proto-tektites, impact spherules, impact glass breccia, and other materials display Zn loss from volatilization that augments the Zn isotope composition of the residual phase (i.e., increases $\delta^{66}Zn$ in the melt droplet), thus acting as a distinct mechanistic indicator of Zn loss via volatilization.

While other mechanisms cannot be entirely ruled out, it is most logical and probable that the observed high $\delta^{66}Zn$, low Zn

concentration trend acts as a multi-dimensional mechanistic tracer of Zn evaporation at the K–Pg boundary. Considering Zn volatilization as the operative process, the data can then be further scrutinized to elucidate details of material mixing during impact and sedimentation. It can also be used to compare the observed Zn volatilization trend (higher $\delta^{66}$Zn with lower Zn concentration) with that for similar—though not identical—datasets in the literature, namely tektites.

**Binary mixing systematics and a distillation model of K–Pg boundary Zn isotope compositions.** Zinc concentrations derived from the various depositional environments vary. The carbonate marine depositional setting of the Caribbean possesses the lowest Zn concentration at 4–8 ppm[31,48]. The terrestrial K–Pg boundary settings retain higher Zn contents (70 ppm)[49], and those from mudstone siliciclastic marine environments have the highest Zn contents (200 ppm). Correspondingly, Zn isotope compositions at the K–Pg boundary also vary. The measured K–Pg boundary Zn isotope compositions of bulk rock values obtained from the Caribbean (marine), Montana, and Missouri (terrestrial and transitional, respectively) have higher $\delta^{66}$Zn values relative to typical continental crustal (terrestrial) compositions (Fig. 2).

In the most parsimonious terms, bulk K–Pg boundary sedimentary layers represent an admixture of un-volatilized "normal" sediment (authigenic and exogenic) and partially volatilized impact ejecta. While not quantitative, simple binary mixing relationships between these two end members (target rock and partially volatilized ejecta) provide useful insights. Modeling and discussion herein focus on tektite compositions as the high $\delta^{66}$Zn, low Zn concentration end member proxy. Other fallout material could possess this volatilized signature; however, tektites are the most studied/representative proxy materials available to model. Binary mixing models (Fig. 3A) define two apparent mixing trends that have a common, low Zn concentration, high $\delta^{66}$Zn end member defined as (1) a devolatilized impact fallout material composition with lower Zn concentration (1 ppm) and a higher $\delta^{66}$Zn value ($\delta^{66}$Zn ~1.8‰), modeled after average tektite compositions[31]. The two crustal mixing end members are: (1) carbonate with ~8 ppm Zn and ($\delta^{66}$Zn of +0.37‰; i.e., the basal sample in the ODP Leg 165 drill core), and (2) a terrestrial crustal composition with higher Zn concentration (70 ppm) and lower $\delta^{66}$Zn (+0.27‰) (Fig. 3A). The two mixing trends presented on Fig. 3A indicate that in the transitional and terrestrial K–Pg boundary sediments, ~10% of the Zn present is derived from partially volatilized ejecta material. In contrast, the marine carbonates indicate a larger fraction of the volatile Zn signature because the background concentrations of Zn are significantly lower, resulting in a less contaminated signature of the fractionated ejecta mass (a higher signal-to-noise ratio). These mixing lines, especially their slope (a function of the magnitude of fractionation per unit Zn lost), very closely mirror the model for Zn isotope fractionation in the melt sheet and crater fill sediments of the Sudbury impact[19]. While binary mixing using tektites as a proxy for impact ejecta material is of course a simple approximation, the model results—and the way that the compositions track the impact process—strongly indicate the incorporation of partially volatilized material from a large bolide impact. While it is not possible to constrain endogenous vs. exogenous material in the K–Pg sedimentary layers at present (and thus also the finite extent of volatilization), the roughly similar concentrations of Zn and the estimated 10% of Zn from partially volatilized material (i.e., from impact ejecta material), is well within the range of ~7–20% as reported in the literature[6,13,50,51].

The qualitative nature of the modeling relates to the large range of tektite compositions observed (e.g[16,47].), and their use as a proxy stand-in for bolide ejecta. Consequently, there is a large range of potential impact spherule Zn isotope compositions, and

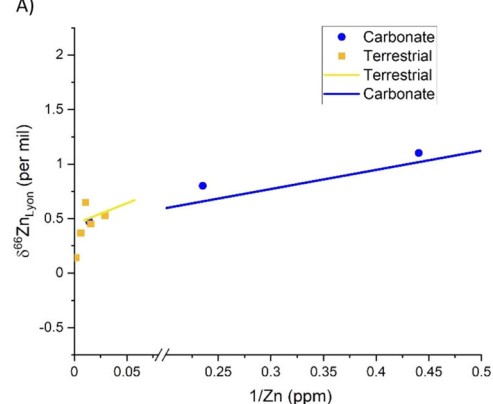

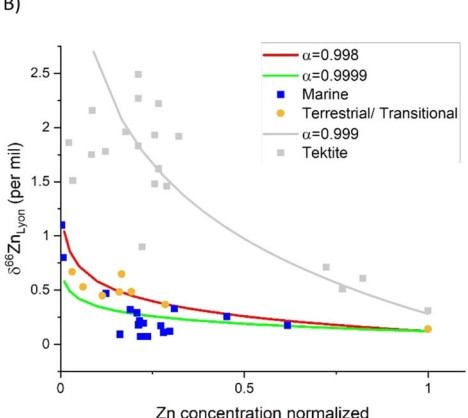

**Fig. 3 Modeled Zn isotope data by distillation. A** Zn isotope data from the carbonate and transitional/ terrestrial environment showing two different mixing trends. Larger portions of the volatilized Zn signature are found in the lower Zn concentration data. **B** Zn isotope composition ($\delta^{66}$Zn) vs. normalized Zn concentration for literature tektites and samples from K–Pg boundary sedimentary layers. Zn concentrations have been normalized such that data can be directly compared. Gray squares represent literature data for tektites[47]. The gray dashed line represents the theoretical Rayleigh distillation curve for $\alpha = 0.999$ (empirical fractionation factor) with an initial $\delta^{66}$Zn of 0.28‰ (BSE). Blue circles represent sample data for shallow and deeper water marine sites (Mississippi and New Jersey, respectively), and green triangles represent sample data for terrestrial and transitional sites (Montana and Missouri, respectively). The red dashed lines represent the theoretical Rayleigh distillation curves for the effective fractionation factors, $\alpha_e = 0.9998$ and $\alpha_e = 0.9999$ (top and bottom, respectively) with an initial $\delta^{66}$Zn of 0.12‰ (marginal marine environment;[32]. Complementary tektite data of Wimpenny et al.[20] are encompassed by that of Moynier et al.[47], and data from Rodovská et al.[17] lie well above the graphic window due to as-of-yet unexplained high $\delta^{66}$Zn values (up to 3.7‰). These data are acknowledged here and within the canon, but for illustrative purposes have not been included in this figure.

it is not currently feasible to constrain other variables such as changes in local P–T environs during spherule drag heating, variability in initial spherule Zn isotope composition, back-condensation, extent of volatilization, and so forth. Nonetheless, it is viable to compare the Zn isotope data herein to that in the literature using Rayleigh distillation modeling, in order to confirm Zn volatilization as the key process behind the range of Zn isotope compositions observed (Fig. 3B). Using the tektite data of Moynier[48] as representative (legitimate given the data overlap of Wimpenny[20], together comprising most available data), an

empirical and generalized isotopic fractionation factor for tektites can be approximated: $\alpha \approx 0.999$ for $\delta^{66/64}$Zn in ‰. This value acts as a generalized metric for the increase in $\delta^{66}$Zn accompanying Zn volatilization under terrestrial conditions. For data within the current work, an empirical range in $\alpha$ that encompasses the data within the current work (an $\alpha$ envelope) has been determined as $0.99975 < \alpha < 0.99995$ (Fig. 3B).

The empirical $\alpha$ for data from tektites[17,20,47], and the $\alpha$ envelope determined for data within the current work, both imply Zn isotope fractionation from volatilization below that theorized by pure Rayleigh distillation ($\alpha \approx 0.985$ in a vacuum). In all cases, the lower $\alpha$ is likely attributable to diffusional lag[16,47] and/or suppression of Zn isotope fractionation during large-scale evaporation under pressurized (i.e., non-vacuum) conditions[18,20,21]. Specific to data within the current work, with an $\alpha$ envelope well below that for tektites[47], there is added attenuation, most likely due to the ~10:1 signal dilution generated by the ejecta mixing with endogenous materials in the K–Pg sedimentary layers (assuming ~10% Zn from ejecta). Lastly, the signature of volatilization in the K–Pg sediments was further attenuated by the addition of Zn from non-volatilized sources during sedimentation and lithification.

The Zn isotope compositions of the K–Pg sedimentary layers closely adhere to a Rayleigh distillation curve whereby $\delta^{66}$Zn increases as Zn concentration decreases. This striking observation pinpoints that partial Zn evaporation from impact-generated melt spherules is the operative process that caused the unique Zn isotope and concentration data of the K–Pg boundary.

**Other considerations and caveats implied from the Zn isotope signature**. The two siliciclastic marine sites (Mississippi and New Jersey) have elevated Zn isotope compositions at the K–Pg boundary, but their Zn isotope compositions are within the error of the Zn isotope compositions of the continental crust, making this correlation statistically tentative. The Zn concentrations of marine sedimentary layers is on average two or more times greater (150 ppm) than that in terrestrial sedimentary rocks (70 ppm) in North America during the Cretaceous[49]. Therefore, the attenuated Zn isotope excursion in these environs is diluted by the authigenic (endogenous) sedimentary Zn budget. As such, no clear mixing relationship for Zn isotope composition and Zn concentrations in the marine settings exists. Because the location of the K–Pg boundary at the marine sites is known, the data at these locations demonstrate that the mixtures of impact-sourced Zn vs. marine Zn vary within clay intervals and suggests that the two sources mixed syndepositionally, which is consistent with the presence of mass flow, tsunamis, and general marine turbulence directly following the K–Pg impact[6].

At present, it is not possible to differentiate Zn volatilized from target carbonate or gneissic basement rock, and/or from the bolide, as the Chixculub bolide struck an area of 2–4 km thick limestone overlying gneissic basement rock. Organic materials preserved within the limestone unit could potentially possess higher Zn isotope values[30], but the concentrations of Zn in marine carbonates are significantly lower (~1 ppm[52]) than that in compositional candidates for the bolide (~13–800 ppm[37]) or in gneiss (~40 ppm). Although carbonate is easier to volatilize, it probably contributes <1% of the Zn to the overall mass balance that defined the Zn isotope composition of impact ejecta (e.g., type 1 spherules).

**The implications of the mass evaporation event and Zn mass balance**. Summarizing the above, the $\delta^{66}$Zn vs. 1/Zn concentration trend observed in the K–Pg boundary layers, though consistent with the general process of impact-induced volatilization of Zn, is markedly suppressed relative to that expected from

Rayleigh distillation alone. For instance, there is less Zn isotope fractionation associated with Zn loss than expected from a model based purely on evaporative loss[47]. By mass balance, if impact ejecta (mainly spherules) became enriched in isotopically heavy Zn, it follows that isotopically lighter Zn must exist somewhere in the rock record, presumably subsequent to the isotopically heavy layers. The ODP Leg 165 Sample 1001-38R 13–15 cm, at the top of the Caribbean site, has the lowest Zn isotope value in the dataset by a significant margin (lower in fact than the typical range for sedimentary rocks). Though speculative (due to a lack of corroborative evidence), this outlier could represent later fall-out from the same volatilized Zn reservoir that is complementary to the isotopically heavy Zn that characterizes the earliest fallout material. Thus, the light Zn isotope reservoir required for mass balance may reside within the boundary sequences, and future work may assist in locating/identifying this complementary reservoir. While certainly a data-limited and therefore notional interpretation, this is further supported by the occurrence of tektites with light Zn isotope compositions where back-condensation has been identified as the operative mechanism[53].

Future investigations of K–Pg boundary Zn isotopic compositions of fallout products, as well as for adjacent layers where complementary isotopically light Zn may reside, will enhance our understanding of the Zn isotopic anomalies found at these K–Pg sites. Moreover, the observations of the current work, namely the utility of Zn isotopes as a mechanistic fingerprint of impact-related volatilization, call for the investigation of Zn isotope compositions within other sedimentary layers containing impact fallout in the rock record.

## Methods

**Sampling strategy and characterization of sedimentary impact layers**. The sampling strategy of boundary sediments and surrounding sedimentary layers varied by location and material availability. Four to five samples were collected in the boundary layer at the Mississippi, ODP Leg 165 (site 1001A&B), and New Jersey locations (Fig. S1). Only one sample was obtained at the boundary at each of the Montana and Missouri locations. In these latter locations, samples encompass a broader stratigraphic interval.

All sections chosen have K–Pg boundary characteristics identifiable by paleontological and geochemical criteria. For instance, Ir anomalies have been reported along with PGE concentrations and interpreted as impact-related for the Montana and New Jersey sites, and type 1 and 2 impact spherules (splash-form melt ejecta and condensates, respectively[11]) exist in all locations, as does the distinctive K–Pg ash layer. The Missouri location is unique, as brecciated samples clearly contain clasts of spherules, demonstrating that this location has texturally reworked materials.

Two sampling tactics were employed to observe both micro- and macroscopic Zn isotope signatures across the boundary. For the Mississippi, ODP Leg 165, and New Jersey sites (ODP Leg 174AX site Bass River), four samples from within the boundary clay layer, along with several samples within 40 cm of the boundary were selected. Limestone from the base of the boundary sequence of the ODP Leg 165 site 1001 section was selected to provide a representative (though statistically limited) Zn isotope signature of the carbonate platform prior to impact[54]. This sample clearly contains some ash, but no spherules are reported. For each interval sampled, ~1–8 g of rock material was powdered for isotopic analysis. Visible type 1 spherules (Supplementary Fig. 1) from the Mississippi marine samples were hand-picked for phase-specific Zn isotope analysis. All samples were powdered to silt size (<63 μm) or finer prior to chemistry to ensure efficient and complete digestion.

Approximately 200–300 mg of sample was dissolved in a two-stage process. First, a 6 ml mixture of ultrapure 10M HF +8M HNO$_3$ was heated in 60 ml polyfluoroalkyl beaker and the solutions were dried. A second mixture of ultrapure aqua regia was used and complete digestion of the samples was visually confirmed. The Zn in post dried salts was purified using anion exchange resin (BioRad MP-1, 200-400 grain size, HCl form) under the protocol defined by [55]; the purification protocol was done twice to ensure complete removal of matrix elements. The samples were measured by multi-collector inductively coupled plasma mass spectrometry on a Thermo Neptune Plus at Rutgers University and Pennsylvania State University. Run conditions are identical to those reported in[56] with the instrument in low resolution mode. Sample solutions of ~200 ppb Zn were doped with 100 ppb Cu (NIST 976) to correct for mass bias with the exponential law[55]. The samples were further corrected with standard-sample-standard bracketing using the AA-ETH Zn isotope[57]. Data have been reported relative to JCM Lyon standard (which is no longer available) by applying a +0.27‰ difference[58–60]. The

error of the standard compared to itself throughout the four session is ±0.05‰ and is considered a conservative estimation of error as duplicate analyses of the same solutions and complete procedural duplicates of the USGS BVHO-2 rock standard (+0.28 ±0.03‰, 2σ, n = 8, overlapping values found in[61]) are significantly less.

Zn concentration measurements were conducted via quadrupole ICP-MS, and calculated gravimetrically using 100 mg aliquots of the same powdered samples. The ICP-MS instrument conditions and data reduction protocols are found in[62]. BVHO-2 Zn concentration were within 8% of reported values. Scanning electron microscope (SEM) images were obtained on a JEOL 6460 SEM at Juniata College (Supplementary Fig. 1).

## Data availability

The authors declare the main data supporting and used by the study are available within the article and its Supplementary Information documents. Related/extra data files are available from the corresponding author upon request.

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

## Acknowledgements

We appreciate the support for the project from NSF grant proposal no. 1924177. Kelly Finan's assistance in figure creation is also greatly appreciated.

## Author contributions

R.M. and M.S. jointly conceived this idea. M.S., M.G., G.P., L.G., F.E.O-I. and N.L. kindly provided the materials for analysis. R.M. and L.G. measured the samples. B.M. and S.-A.L. developed the idea and provided key insight into the comparative analysis and modeling. R.M. wrote the document with significant contributions from all authors. All authors contributed to drafting the document.

## Competing interests

The authors declare no competing interests.
