## [Peer Review File · Nature Communications]

REVIEWER COMMENTS

Reviewer #1 (Remarks to the Author):

Review of Mathur et al. "Fingerprinting the K-Pg boundary impact with Zn isotopes"

Balz Kamber, January 2021, Brisbane

Mathur et al. present new stable Zn-isotope values for sedimentary layers of the K-Pg boundary from four locations (terrestrial to marine) in variable proximity to the established Yucatan peninsula Chicxulub impact structure. By analysing sediment from below and above the actual K-Pg layer, the authors were able to identify isotopically heavy Zn in the terrestrial sites (where the local Zn concentration is low) outside analytical uncertainty and within the marine sections, elevated isotope compositions but at a more diluted level due to the higher local Zn concentration in the sediment.

The authors interpret their data with a Rayleigh distillation model of preferential loss of light-over-heavy Zn from superheated melt either on impact and/or transport of the melt with additional potential for condensation of heavy Zn into condensate-type spherules. This is the first demonstration, to my knowledge, of direct impact-induced stable metal fractionation at the K-Pg boundary (but see Gilleaudeau et al., 2018 for other indirect stable isotope perturbations). From my limited analytical understanding, the data are sound, the topic suitable for the journal and the findings potentially sufficiently significant to warrant publication in the journal.

Although I enjoyed reading the manuscript and find the science exciting, my impression is that in its current form, the manuscript is not sufficiently focussed to be of broad enough appeal to a general readership. It would have been nice to have had line numbers to refer to and I wonder whether this is a m/s that was not specifically prepared for the journal but instead transferred from a different Nature publication (e.g. I believe the journal permits 70 references and more illustration material?). If the methods were moved and the full length, references, illustration materials permitted by the journal were utilised, I think a revised m/s could make for a very interesting story.

Areas that could be improved:

i) The question of the fate of the isotopically light Zn?

While the K-Pg boundary continues to sustain a stimulating scientific debate about bolides and extinction, there is a more general importance to large impacts and element transfer during bolide collisions. This arises from the fact that the bulk of the material that forms the Earth has been aggregated by collisional growth. Substantial uncertainty still exists about the behaviour of volatile (e.g. H₂O) and moderately volatile elements. There is new evidence that this behaviour depends on atmospheric pressure, redox-state and the nature of the liquid (silicate vs. pure elemental). Ultimately, the question relates to whether the Earth captured its volatile inventory as a late veneer or whether the Earth's C, H₂O, etc. are endogenous (e.g. Sossi et al., 2018).

Zinc isotopes may be one tool to advance this knowledge as Zn behaves moderately volatile and preferentially loses light isotopes during volatilisation. If the mass of target and bolide are conserved on a global scale during a very large impact event, then mass balance dictates that if the target is left depleted in light Zn, some of the ejecta (late condensates) would be expected to have light isotope compositions and the effect of bolides would be a mere re-distribution. I believe this was the hypothesis proposed by Albarede et al. (2013). If, by contrast, some volatiles are lost to space, e.g. by hydrodynamic escape (in subaqueous impacts, such as Chicxulub), a mass imbalance occurs, which could lead to non-chondritic terrestrial isotope compositions. For the most volatile elements, impacting could thus even lead to a net loss rather than gain (see. e.g. Kamber&Petrus, 2019). It may well be that the authors have considered this issue and that the contribution of light Zn might be immeasurable (or expected at more distant field sites) but I feel that not discussing this more general context is an omission that is difficult to justify.

ii) The nature of the Chicxulub target

For a broader readership, I'm missing an upfront description of the impact target geology. A half-submerged carbonate platform, rich in organics and sulphate, underlain by a crystalline basement. The authors appear to suggest that since carbonate is generally low in Zn concentration and because the organics (oil, kerogen,

etc.) are low in total abundance, the Zn they are seeing in the ejecta layers is largely derived from the basement. I'm somewhat familiar with the literature about the shock melts from the drill cores into the Chicxulub structure and would have thought that the composition of the studied spherules (are these mainly silicates and oxides?) could be used to estimate the relative proportion of silicate/carbonate. On the topic of carbonate and Zn, I found it a bit misleading to state that carbonates are generally low in Zn because carbonates can also host very significant Zn-mineralisation, particularly in basins where there is an excess of S. I would like this aspect of the study site better explained and explored.

iii) Is there any sign of the bolide?

The authors argue that the heavy Zn-isotope composition in the K-Pg boundary sedimentary rocks is not directly derived from an isotopically heavy bolide (CM2 type chondrites). While I do not disagree with this conclusion, I think the claim needs further firming up with calculations of expected elemental Cr/Ni (e.g. Frei&Frei, 2002) vs. Cr/Zn and Cr-isotope (e.g. Gilleaudeau et al., 2018) vs. Zn-isotope ratios.

Minor details:

The authors use the word 'proximal' to describe the distance of a place from the impact site but also the stratigraphic distance of a sedimentary rock bed/layer from the impact ejecta. I think this creates unnecessary confusion and considering the topic of the manuscript, the term should here be reserved for physical distance from impact site.

As alluded to above, I would like to see a histogram of Zn-isotope compositions and concentrations (or maybe Zn concentration vs. isotope composition?) of marine sedimentary rocks and associated minerals. I also miss more reference to the lunar Zn-isotope data, keeping in mind that the Moon has no atmosphere and therefore, evaporative loss there is almost certainly different than on Earth (e.g. Sossi et al., 2019).

References

- Albarede, F., Ballhaus, C., Blichert-Toft, J., Lee, C.T., Marty, B., Moynier, F. and Yin, Q.Z., 2013. Asteroidal impacts and the origin of terrestrial and lunar volatiles. *Icarus*, 222(1), pp.44-52.
- Frei, R. and Frei, K.M., 2002. A multi-isotopic and trace element investigation of the Cretaceous-Tertiary boundary layer at Stevns Klint, Denmark-inferences for the origin and nature of siderophile and lithophile element geochemical anomalies. *Earth and Planetary Science Letters*, 203(2), pp.691-708.
- Gilleaudeau, G.J., Voegelin, A.R., Thibault, N., Moreau, J., Ullmann, C.V., Kläbe, R.M., Korte, C. and Frei, R., 2018. Stable isotope records across the Cretaceous-Paleogene transition, Stevns Klint, Denmark: New insights from the chromium isotope system. *Geochimica et Cosmochimica Acta*, 235, pp.305-332.
- Kamber, B.S. and Petrus, J.A., 2019. The influence of large bolide impacts on Earth's carbon cycle. *Elements: An International Magazine of Mineralogy, Geochemistry, and Petrology*, 15(5), pp.313-318.
- Sossi PA, Nebel O, O'Neill HS, Moynier F. Zinc isotope composition of the Earth and its behaviour during planetary accretion. *Chemical Geology*. 2018 Jan 20;477:73-84.
- Sossi PA, Klemme S, O'Neill HS, Berndt J, Moynier F. Evaporation of moderately volatile elements from silicate melts: experiments and theory. *Geochimica et Cosmochimica Acta*. 2019 Sep 1;260:204-31.

Reviewer #2 (Remarks to the Author):

Mathur et al., Nature Communications. Jan-2021

General comments

I was interested to review this manuscript concerning the behaviour of Zn isotopes in sediments from the K-Pg boundary. Having some experience with moderately volatile element isotope systems I thought that this would be a novel and fairly high profile study that would be of broad interest to the readers of Nature Communications. However, there are numerous problems with the paper and I cannot recommend publication in this or any other journal until they are addressed. The main problems are as follows:

Zn data: I'm sure the data is of high quality but the figures and text do not provide any way to establish whether the 'significant' Zn isotope fractionation in the boundary sediments is really that. Based on Figure 1 it looks like there is a consistent positive spike in the boundary sediments but maybe not outside of uncertainty. The fact that the boundary sediments have Zn isotope ratios that are similarly heavy (+0.4-0.5permil) to some weathered sediments in the literature is surely an important consideration when

interpreting the data. Instead, this is conveniently ignored.

Writing: In general the writing is poor, there is an alarming lack of references, its incredibly vague in places. No data is provided to back any statements up.

Inconsistencies: The authors do not explain why the type-1 spherules have relatively unfractionated $\delta^{66}\text{Zn}$ values of +0.4 whereas they assume the impact derived material has a $\delta^{66}\text{Zn}$ value of +1.4 in the mixing calculation. The authors assume that vapor condensates should be isotopically heavy (citing Toutain et al., 2010) but that is not true. In fact Toutain et al., (2010) said the opposite.

Figures: They do not add anything useful to the study. Figure 1 doesn't show the actual data, just places a best fit line through it. No uncertainties are shown. Figure 3 is supposed to show some kind of mixing line but that appears to have been left out.

Overall Significance: Even if a volatile depleted signature could be identified at the K-Pg boundary what else can Zn isotopes provide about the processes that occurred or the composition of the impactor or target material. Based on this, not much.

Overall, it's a nice set of data that is presented poorly, not well explained and the author needs to read and cite more of the relevant literature. Its not good enough for most journals, including Nature Communications.

Page by page comments:

General – it would be great to have line numbers here. For both the reviewer and for the authors trying to figure out what I am commenting on. I have definitely forgotten to provide them before, but the journal usually enforces it.

Page 3:

Discern clear geochemical signatures of what?

The fact that the K-Pg sediments contain high Ir and anomalous isotopic compositions (i.e. Cr54) are pretty good indicators of an extraterrestrial impact, rather than just a simple indicator of change. I think I understand what is being said but perhaps it needs to be worded differently. The idea is, I guess, that we have no information from the K-Pg sediments about the physical processes associated with such an impact.

Page 4:

There needs to be a better introduction as to why metal stable isotopes would be important. Who cares if no data currently exists? Its not a useful lead off statement. You should introduce the moderately volatile elements and why they can provide useful information about physical processes that would, in turn, be relevant to interpreting the K-Pg sediments.

Although igneous rocks have a tight range of Zn isotope ratios there is evidence that weathering and weathered sediments can have more diverse isotopic compositions. E.g. Lv et al., 2016; Little et al., 2019...). Heterogeneity in the Zn isotope ratio can be imparted by weathering, which could be important when interpreting the K-Pg sed data.

Final paragraph: Zn isotopes are sensitive to volatility driven processes but are not simply controlled by them. Other processes such as mineral-melt partitioning in magmatic systems (e.g. Sossi et al., 2018) or weathering (Little et al., 2019) can fractionate Zn.

Tektites have been analysed by several studies not referenced here: Moynier et al., 2009; Rodovska et al., 2017; Wimpenny et al., 2019...there are probably more as well!

Page 6:

The instrument is a MC-ICP-MS and the acronym should be defined.

Second paragraph: This section is poorly written and should be extensively edited

Rather than referencing another paper please explain the methods to obtain Zn concentrations fully either here or in a supplemental section. In fact, it might be worthwhile moving some of isotope ratio methods to a supplemental as well.

Page 7:

What is significant fractionation of Zn? This means different things to different people.

What does falling in this category mean?

Why would these layers be inversely correlated?

Is there any indication that leaching of heavy isotopes from other layers would occur? Reference a study that shows this here.

Replace too with to

There needs to be some quantification of Zn fractionation in weathered continental sediments for context. Provide numbers and references. I have seen data that shows $\delta^{66}\text{Zn}$ values of >0.5permil in weathered

sediments (Yv et al., 2016 etc...) This first paragraph is far too vague.

Why don't you start the second paragraph by saying that the K-Pg sediments have higher $\delta^{66}\text{Zn}$ values than strata above and below?

What is the significant uptick? Provide numbers here. This is far too vague. Figure 1 is not very clear – is the uptick outside of uncertainty of the technique?

Again, paragraph 2 suffers from being too vague. I don't know that significance of the changes in Zn isotope ratio from the text or the figure so its hard to evaluate the following interpretation.

Paragraph 3: Do you mean the depositional environments in this study or in general? Where do these numbers come from?

Why should the Zn isotope ratio at the boundary vary as a consequence if fractionation is imparted by evaporative loss? Unless you are talking about the addition of fractionated material from the debris cloud (i.e. two-component mixing).

Page 8:

Figure 2 doesn't show typical crustal compositions so we can't compare the data. Unless CC represents the crust? But that acronym is typically used to denote carbonaceous chondrites...

Paragraph 2: Tektites are not sediments from the K-Pg boundary. Sediments from the K-Pg boundary have high Ir and Cr isotope anomalies that derive from the impactor (as stated in the introduction). As far as I know these have not been found in tektites previously. There must be estimates for the amount of impactor material required to generate these anomalies in the K-Pg sediments. Provide them here.

I don't think that you have proven that the Zn fractionation could not be a signal from the impactor without providing a mass balance calculation based on the amount of impactor material in the K-Pg sediments.

Paragraph 3: You have already mentioned tektites so why are you defining them halfway through the manuscript?

The second sentence is poorly written. What you mean is 'Samples derived from impact events such as...are relatively depleted in Zn and have relatively heavy $\delta^{66}\text{Zn}$ values, indicating loss of Zn by volatilization.

Page 9:

The mixing approach is sensible but Figure 3 does not help the discussion. It doesn't illustrate mixing at all. Where are the mixing lines that are described?!

How does this compare with other estimates of impactor material in the K-Pg sediments?

Paragraph 3: A great deal of the tektite references are missing here.

The data in S2 do not comply with a simple Rayleigh distillation curve. They are highly scattered and the data appear to sit on different curves. The assertion that these data are better than the available tektite data is completely unfounded and not very scientific. Moyniers data appears more convincing based on Figure S2. Please also see Wimpenny et al., (2019) who also provide a well-defined Rayleigh fractionation relationship in tektites.

Page 10:

Paragraph 1: This is a highly speculative interpretation of the alpha values given the scatter in the data.

The problem with interpreting tektite data is that the precursor material is not well constrained, particularly for australites which form the majority of current sample sets. This makes the alpha for tektite data relatively imprecise. This should be considered because currently I believe the data are being over interpreted. The best you could say is there may be some similarity in alpha during evaporative loss if ~10% of the K-Pg material derived from volatile depleted impact debris.

Its consistent rather than incredibly concordant.

Where is the Sudbury data from?

I don't think you can say much about the process other than the sediments consist of material that has lost volatiles.

Paragraph 2: This repeats much of what was stated on page 8. If the $\delta^{66}\text{Zn}$ values are within uncertainty of the continental crust can you really state that the marine sites have elevated $\delta^{66}\text{Zn}$ values?

Paragraph 3: Wouldn't impact melt spherules also be the source of heavy Zn in the terrestrial sections as well?

Based on the information in the supplemental the spherules have $\delta^{66}\text{Zn}$ values of +0.39 and +0.43...

If the impact spherules are the source of isotopically heavy Zn isn't there a mass balance issue? The spherules have $\delta^{66}\text{Zn}$ values that are similar to UCC. Therefore to shift the $\delta^{66}\text{Zn}$ value of the boundary layer would require the majority of Zn to be derived from the impact debris, rather than the 10% that is assumed on page 9.

Page 11:

I assume that this Zn data from Sudbury is from Kamber & Schoenberg? Why is this not cited?

But the isotopic value of impact spherules is not elevated compared to sediments from the terrestrial boundary layers...Some of the marine sediments have $\delta^{66}\text{Zn}$ values that are very similar to the impact spherules which would imply that the spherules contribute >90% of Zn to the rock based on the arguments used here. Also, the assumed composition of the impact spherules in the mixing calculation was +1.4 permil. Why do the spherules that were analysed have $\delta^{66}\text{Zn}$ values that are so much lower? There is a lack of consistency here meaning that either I am missing something (which is possible) or the explanation needs to be rethought.

Where does this 100-200ppm estimate come from? Is it from analyses of CM chondrites (e.g. Luck et al., 2005)? What if the organics in the limestone were preferentially volatilized? Could they have contributed disproportionately to the boundary layer?

Paragraph 3: Hasn't this been explained already?

Similar observations have also been made for Ga (Wimpenny et al., 2020)

Page 12:

Paragraph 1 – The key problem is pressure into which evaporation takes place. The theoretical kinetic fractionation factor can only be attained in a vacuum. Read a few papers by Richter for more information on this. Higher vapor pressures suppress the fractionation thus the alpha from volatile loss in terrestrial systems will be closer to 1. There is plenty of data out there and references that could be made. For example, see Yu et al., (2003) who show a relationship between K isotope fractionation and vapor pressure. See associated work in Wang & Jacobsen 2016. Unfortunately, this paragraph makes use of none of the prior work and as a result is not very informative. It reads like the suppression of alpha is a new discovery. It's not.

Paragraph 2: Gas condensates on the Moon are highly enriched in light Zn, as one would expect in a kinetic process, where the forward reaction favors the lighter isotope. Fumeroles are a more complicated system, where the composition of the gas phase also reflects the evolving reservoir. However, if you actually read Toutain et al., (2010) you will see that the fumarolic gases are interpreted to condense light Zn, but the remaining Zn in the gas becomes heavier as the gas continues to ascend. Thus, the first condensates in the vent would be light and then get progressively heavier. Any gas condensate would thus be enriched in light Zn isotopes contrary to the assertion here for type-2 spherules.

Where is the P-T data from?! Why are there no references in this manuscript?!

Figure 1 – Add the data points and uncertainties not just a best fit line. This figure is useless right now as I have no idea whether the 'excursions' are real or within uncertainty of the other data.

Figure 2 – What is CC? What is a gas cond and where are the data from? What tektite data is used here?

Figure 3 – I do not understand how this demonstrates a mixing relationship.

Reviewer #3 (Remarks to the Author):

This is a well-argued and well-written manuscript, with interesting and obviously high-quality data. I have no concerns about the general idea, the data, or the interpretation and conclusions. Most of the questions I asked myself when reading the manuscript were answered somewhere in the manuscript. The authors could strengthen their arguments by maybe rewriting or amplifying some of their points, as follows.

- In the abstract, there are some what I'd say filling sentences or sentence parts. It would be more succinct to clearly point out what the goals of the present work were/are (what is the scientific question that the authors set out to answer); explain what variety of materials was analyzed and why, explain why alteration etc does not affect the Zn isotopic composition, and clearly state even in the abstract that volcanism cannot explain the observed values.

- In general I also thought that the question of (Deccan) volcanism as a possible source - given how a small but vocal minority harps on this topic - should be more clearly dealt with, and in the text more clearly stated why the data do not agree with such an idea.

- The connection to the processes that have been observed in tektites could also be made more clearly; the authors note that in some cases K-Pg spherules were analyzed, but were those glassy or not? Some rudimentary petrographic descriptions (a couple of sentences) of the samples would be useful in the main text.

- I am a bit sad that no data tables are present in the main text. These days too much important material gets pushed to supplements.... if space allows it it would be good to put a table there.

Otherwise the figures look ok, the references appear in order and up to date, so I have no additional comments there.

I look forward to seeing this interesting work published.

Christian Koeberl
University of Vienna, Austria
(Jan. 2, 2021)

Response to Reviewer: Balz Kamber, January 2021, Brisbane

Although I enjoyed reading the manuscript and find the science exciting, my impression is that in its current form, the manuscript is not sufficiently focussed to be of broad enough appeal to a general readership.

To make the document more accessible to a broader audience we did the following:

- 1) Changed the abstract to solely focus on geochemical signatures.*
- 2) We deleted the first two sentences from the abstract and moved these general statements to first introduction paragraph.*
- 3) We added >10 new references and discussion about these references to the introduction and discussion.*

It would have been nice to have had line numbers to refer to and I wonder whether this is a m/s that was not specifically prepared for the journal but instead transferred from a different Nature publication (e.g. I believe the journal permits 70 references and more illustration material?).

Agreed, page numbers added and we also added the suggested references along with others.

If the methods were moved and the full length, references, illustration materials permitted by the journal were utilised, I think a revised m/s could make for a very interesting story.

We moved the methods to the appendix material and moved the data table to the main document.

Areas that could be improved:

i) The question of the fate of the isotopically light Zn?

While the K-Pg boundary continues to sustain a stimulating scientific debate about bolides and extinction, there is a more general importance to large impacts and element transfer during bolide collisions. This arises from the fact that the bulk of the material that forms the Earth has been aggregated by collisional growth. Substantial uncertainty still exists about the behaviour of volatile (e.g. H₂O) and moderately volatile elements. There is new evidence that this behaviour depends on atmospheric pressure, redox-state and the nature of the liquid (silicate vs. pure elemental). Ultimately, the question relates to whether the Earth captured its volatile inventory as a late veneer or whether the Earth's C, H₂O, etc. are endogenous (e.g. Sossi et al., 2018).

Zinc isotopes may be one tool to advance this knowledge as Zn behaves moderately volatile and preferentially loses light isotopes during volatilisation. If the mass of target and bolide are conserved on a global scale during a very large impact event, then mass balance dictates that if the target is left depleted in light Zn, some of the ejecta (late condensates) would be expected to have light isotope compositions and the effect of bolides would be a mere re-distribution. I believe this was the hypothesis proposed by Albarede et al. (2013). If, by contrast, some volatiles are lost to space, e.g. by hydrodynamic escape (in subaqueous impacts, such as Chicxulub), a mass imbalance occurs, which could lead to non-chondritic terrestrial isotope compositions. For the most volatile elements, impacting could thus even lead to a net loss rather than gain (see. e.g. Kamber&Petrus, 2019). It may well be that the authors have considered this issue and that the contribution of light Zn might

be immeasurable (or expected at more distant field sites) but I feel that not discussing this more general context is an omission that is difficult to justify.

Agreed, this is an excellent point. We have measured a lighter Zn isotope composition at the top of the boundary and can account for this general mass balance. The following language was added to the document.

It has been postulated that studies of impact ejecta layers, melt sheets, etc. may provide insight on the possibility of volatile element loss from Earth during large impacts such as that at the K-Pg and Sudbury¹. The Earth is depleted in many volatile and moderately volatile elements relative to potential source materials (e.g. chondritic meteorites²). However, the heavy Zn isotope enrichment in the K-Pg sedimentary layers of the current work relative to estimates of the bulk Earth Zn isotope composition from 0.16-0.28 ‰³⁻⁵, coupled with the observation that Zn isotope vs [Zn/Mg] systematics for Earth and the meteorite record follow a trend near-completely juxtaposed to that which would indicate loss by volatilization³ (even when considering core-mantle differentiation and partial melt), indicates that large bolide impacts do not result in significant Zn loss from Earth's atmosphere. This is in stark contrast to the volatility-related trend seen for the Moon due to outgassing or its own formation by impact⁶⁻¹⁰. This means that from a mass balance point of view, Zn isotopes were re-distributed during impact events on Earth (and not modifying bulk Earth Zn elemental or isotope compositions). Therefore, isotopically lighter Zn must exist in the fallout layers of the rock record. The ODP Leg 165 Sample 1001-38R 13-15cm, at the top of the Caribbean site, has the lowest Zn isotope value in the dataset by a significant margin (lower in fact than the typical range for sedimentary rocks). This outlier could represent condensation from the volatilized Zn reservoir towards the end of the impact event (after condensation and ballistic ejection, and thus depleted in heavy Zn isotopes). The stratigraphic positioning and lower total Zn concentration of the sample indicate that materials with lighter Zn isotope compositions exist at the top of the boundary, associated with deposition towards the end of the fallout event. Thus, the light Zn isotope reservoir required for mass balance resides within the boundary sediment. While certainly a data-limited interpretation, this notion is further supported by the occurrence of tektites with light Zn isotope compositions where back-condensation has been identified as fractionating mechanism¹¹.

Along with the above, at relevant points within the revised manuscript, we have further discussed co-eval processes such as condensation.

We are familiar with the work of Sossi et al. (2019), and other publications with this focus. While we fully agree with the Reviewer, especially as regards the importance of understanding the history and fate of Earth's volatile elements and the utility of isotopes in scrutinizing such questions, we believe that such an endeavour is far outside the focus (and purpose) of the present work. The present study is aimed at understanding the K-Pg boundary (and its cause) through the lens of Zn isotopes. Such an inquest would also require understanding heating of the impactor itself upon entry, and/or the impactors prior heating history, as these factors could cause substantial isotope fractionation as well (e.g. Mahan et al., 2018 – A history of violence...). Ascertaining answers to the questions posed by the Reviewer would require an immense dataset from a large array of bolide impacts, coupled to relevant experimental work and numerical modelling (e.g. thermodynamics, and physico-chemical) that is beyond the main purpose and scope of the current study. That said, such efforts should be at the forefront of future research

endeavours.

ii) The nature of the Chicxulub target

For a broader readership, I'm missing an upfront description of the impact target geology. A half-submerged carbonate platform, rich in organics and sulphate, underlain by a crystalline basement.

The authors agree, and we have changed section title, and have completely reorganized and rewrote this section to address these concerns:

Evidence for a distinct volatile Zn isotope signature in boundary layer sediment

The Zn isotope composition of the K-Pg layers analyzed possess higher Zn isotope values than the surrounding sedimentary rocks (Figure 2). The Zn isotope composition of the boundary layer does not reflect that of the target rocks which the bolide struck, or material from the bolide itself. It has been well documented that the impact struck a carbonate platform (with some layers rich in organics and sulfate), underlain by a metamorphic crystalline basement¹². The highest Zn isotope value recorded in the K-Pg boundary exists in the Caribbean site ($\delta^{66}\text{Zn}=+1.1$ per mil). This value is greater than any whole rock value reported for sedimentary rocks to date (highest values reported in¹³ at $\delta^{66}\text{Zn}=+0.78$ per mil in a sulfide rich rock and $\delta^{66}\text{Zn}=+0.42$ silicate whole rock lacking sulfide¹⁴, Figure 2). The only rock materials on earth and from extraterrestrial materials with $\delta^{66}\text{Zn}$ values that can be sufficiently high are hydrothermal sulfide ores^{15,16} and tektites^{10,17,18} (melt materials derived from the target rocks of meteorite impacts), with only few examples existing in the meteorite record⁴. Isotopically, none of the target rocks (no sulfide rich Zn ore deposits or sulfides are known to occur in the K-Pg target limestone sequences), or samples from the meteorite record, possess Zn isotope compositions similar to the highest values presented here in the boundary sediments (Table 1).

Interpretation of the origin of the distinctive, heavier Zn isotope compositions of the K-Pg boundary layers analyzed relates to several different physical processes such as: melt-induced fractionation, inherited signatures from the target rocks where the meteor struck, meteoritic materials that settled within the boundary, secondary alteration/depositional processes, and residuum from volatilized metal from the impact event that settled in the "fall-out" layers. Geochemical signatures related to these processes have been identified for the first four of five processes listed with both radiogenic isotope and concentration data in many different studies using similar sampling methodologies presented here¹².

Mechanisms relating to the melting of target rocks at the impact site do not generate elevated Zn isotopes values present in the boundary layer. Multiple studies of Earth's mantle melts and their associated peridotite residues reveal that Zn does not routinely fractionate greater than 0.2 per mil^{3,19-22} during mantle potential temperatures of 1300 to 1500°C. Considering that the instantaneous melting temperature associated with impact likely exceeded this range, the production of melt upon impact is unlikely by itself to produce the largest Zn isotope fractionation measured in the Caribbean site samples.

The target sources rocks, the impactor itself and secondary migration of Zn post deposition also do not explain the unique Zn signature in the boundary layer. Extensive study of Zn isotope compositions in Earth materials over the last 20 years indicates that the vast majority of terrestrial

geologic processes do not significantly fractionate Zn isotopes, with the exception of evaporation. Redox and other kinetic reactions associated with biogeochemical interactions in soils and shallow depositional environments produce relatively small Zn isotope fractionations (0.1 to 0.2‰^{13,14,23,24}). Measured Zn isotope fractionation among geochemical products and reactants rarely approach or exceed 0.3 ‰ in ore minerals and clays^{15,25-27}. While certain mineral and gas phases, and biologically-derived materials, may possess Zn isotope compositions outside the range reported above (e.g. $\delta^{66}\text{Zn}$ values ranging from -1 to +2 ‰), bulk rock values containing such phases typically do not deviate from bulk silicate Earth (BSE) and continental crustal values, because these materials are often only present at trace levels at the macro-/regional scale. This is most relevant for biogenic carbonates that can have slight enrichments in heavy Zn isotopes²⁸, however these typically contain only ~1 ppm Zn²⁹, while isotopically lighter (lower $\delta^{66}\text{Zn}$) continental margin and basin marine sediments (e.g. $\delta^{66}\text{Zn}$ ~0.12 ‰) can contain 10s to 100s of ppm Zn²⁴.

The stratigraphic changes in Zn isotopic signature of the K-Pg boundary sediments (Fig 1) also indicates minimal potential for terrestrial alteration as a cause for the higher values at the boundary. Significantly, adjacent sediment layers in the sections sampled show no element-isotope mixing systematics. This indicates that no significant leaching of heavy Zn isotopes from under or overlying rock layers took place, mitigating against this process as a cause of $\delta^{66}\text{Zn}$ enrichment within the boundary layer itself, and highlights the otherwise typical nature of the stratigraphic sequence with regards to Zn isotope composition. These observations agree with studies that have addressed secondary fractionation processes in sediments, that indicate such process do not fractionate Zn isotopes in a systematic way or too any large degree^{15,25,30}.

The most practical explanation for the heavy Zn isotope signature at the K-Pg boundary originates from partially volatilized Zn generated during impact. Research on tektites - fused glasses from cooling of impact induced melts¹⁷ - along with complementary work on fused silicates from nuclear blast sites, and experimental Zn volatilization, show that Zn isotopes fractionate by as much as +0.5 to +2 ‰^{1,18,31}. During impact events, ejecta such as proto-tektites, impact spherules, impact glass breccia and other materials display Zn loss from volatilization that augments the Zn isotope composition of the residual phase (i.e. increases $\delta^{66}\text{Zn}$ in the melt droplet), thus acting as a distinct mechanistic indicator of Zn loss via volatilization. The physical composition of such projectiles are often that of the target material mixed with a smaller contribution from the impactor, and therefore the initial Zn isotope compositions of projectiles most likely mimics that of the target material.

Further verification of the volatility related signature for the Zn resides in a comparison to the magmatic Zn isotope signature of the Permian-Triassic boundary (P-T). The P-T boundary possess an abrupt increase of Zn concentration and a concomitant ~0.5‰ decrease in ^{66}Zn occurring ~35 k.y. before the end-Permian mass extinction. The Zn isotopic and concentration changes have been interpreted to demonstrate rapid and massive input of isotopically light Zn from volcanic ashes, hydrothermal inputs, and/or extremely fast weathering of large igneous provinces (LIPs) into the oceans³². From the comparison to the Permo-Triassic boundary, the elevated Zn isotopic ratios and decrease of Zn concentration of the sedimentary rocks in K-Pg boundary sediments are thus inconsistent with the key inputs of isotopically light Zn from processes associated with extensive volcanism. Finally, any input from volcanism and weathering associated with the Deccan Traps, the LIP which precedes the K-Pg boundary³³, would only serve to dampen the positive $\delta^{66}\text{Zn}$ excursion at the boundary—not to generate or enhance it—especially in the marine settings.

The authors appear to suggest that since carbonate is generally low in Zn concentration and because the organics (oil, kerogen, etc.) are low in total abundance, the Zn they are seeing in the ejecta layers is largely derived from the basement. I'm somewhat familiar with the literature about the shock melts from the drill cores into the Chicxulub structure and would have thought that the composition of the studied spherules (are these mainly silicates and oxides?) could be used to estimate the relative proportion of silicate/carbonate. On the topic of carbonate and Zn, I found it a bit misleading to state that carbonates are generally low in Zn because carbonates can also host very significant Zn-mineralisation, particularly in basins where there is an excess of S. I would like this aspect of the study site better explained and explored.

Agreed, we strengthened this document by adding the new section and language above to address these complexities. However, we again note that addressing the complexity of processes and changes in P-T conditions associated with bolide impacts, such that the above question of silicate-carbonate proportions, etc., could be determined, would require building an extensive library of impact layer materials from numerous locales, along with appropriate experimental work and thermodynamic-physico-chemical modelling. This is far outside the scope and purpose of the present work, though certainly merits the attention and efforts of the community.

iii) Is there any sign of the bolide?

The authors argue that the heavy Zn-isotope composition in the K-Pg boundary sedimentary rocks is not directly derived from an isotopically heavy bolide (CM2 type chondrites). While I do not disagree with this conclusion, I think the claim needs further firming up with calculations of expected elemental Cr/Ni (e.g. Frei&Frei, 2002) vs. Cr/Zn and Cr-isotope (e.g. Gilleaudeau et al., 2018) vs. Zn-isotope ratios.

While we agree that inquests such as that noted by the Reviewer are of value, we wish to point out that we only indicate that a wealth of literature suggests only a small fraction of impact spherule material is from the impactor itself, and do not make any argument ourselves. We do not use Zn isotopes or any other observations from the current work to attempt to constrain an answer to this question, as the data do not permit this. We refer to our responses to the previous two questions as further explanation on this.

To further augment our statement in the revised manuscript, and support the notion that these are volatile signatures, we teased out the following passage from the edits requested above:

The Zn isotope composition of the boundary layer does not reflect Zn of the host rocks which the meteorite struck or material from the meteorite. It has been well documented that the impact struck a carbonate platform (with some layers rich in organics and sulfate), underlain by a metamorphic crystalline basement¹². The highest Zn isotope values recorded in the K-Pg boundary exist in the Caribbean site are greater than any whole rock value reported for sedimentary rocks to date (Figure 2). The only rock materials measured on earth and extraterrestrial materials with high values that match these are hydrothermal ores^{15,16} and tektites (melt materials derived from meteorite impacts). Isotopically, none of the target rocks (no Zn ore deposits are known to occur in the limestone sequences) or the meteorites that have been documented possess Zn isotope compositions similar to the highest values presented here in the boundary sediments (Table 1).

Minor details:

The authors use the word 'proximal' to describe the distance of a place from the impact site but also the stratigraphic distance of a sedimentary rock bed/layer from the impact ejecta. I think this creates unnecessary confusion and considering the topic of the manuscript, the term should here be reserved for physical distance from impact site.

Agreed, to clarify this issue, we deleted the term proximal when referring to sedimentary strata and reserved that term for distance from impact structure. Figure 1 was also edited to include arcs as a means to further communicate our ideas.

As alluded to above, I would like to see a histogram of Zn-isotope compositions and concentrations (or maybe Zn concentration vs. isotope composition?) of marine sedimentary rocks and associated minerals

The first Figure 3 had these values listed. However, reviewer was rather unimpressed with this approach. We have edited Figure 3 to display the mixing relationship and the modeled distillation results as a means to accomplish figures with meaning as reviewer #2 requested. We can certainly add a data table with these values or a histogram to the supplemental information. That is simple and if there is room for this data we can provide upon the editors request.

I also miss more reference to the lunar Zn-isotope data, keeping in mind that the Moon has no atmosphere and therefore, evaporative loss there is almost certainly different than on Earth (e.g. Sossi et al., 2019).

All requested references have been added, to this specific request, we added the following:

It has been postulated that studies of impact ejecta layers, melt sheets, etc. may provide insight on the possibility of volatile element loss from Earth during large impacts such as that at the K-Pg and Sudbury¹. The Earth is depleted in many volatile and moderately volatile elements relative to potential source materials (e.g. chondritic meteorites²). However, the heavy Zn isotope enrichment in the K-Pg sedimentary layers of the current work relative to estimates of the bulk Earth Zn isotope composition from 0.16-0.28 ‰³⁻⁵, coupled with the observation that Zn isotope vs [Zn/Mg] systematics for Earth and the meteorite record follow a trend near-completely juxtaposed to that which would indicate loss by volatilization³ (even when considering core-mantle differentiation and partial melt), indicates that large bolide impacts do not result in significant Zn loss from Earth's atmosphere. This is in stark contrast to the volatility-related trend seen for the Moon due to outgassing or its own formation by impact⁶⁻¹⁰.

Response to Reviewer #2 (Remarks to the Author):

Mathur et al., Nature Communications. Jan-2021

General comments

I was interested to review this manuscript concerning the behaviour of Zn isotopes in sediments from the K-Pg boundary. Having some experience with moderately volatile element isotope systems I thought that this would be a novel and fairly high profile study that would be of broad interest to the readers of Nature Communications. However, there are numerous problems with the paper and I cannot recommend publication in this or any other journal until they are addressed. The main problems are as follows:

Zn data: I'm sure the data is of high quality but the figures and text do not provide any way to establish whether the 'significant' Zn isotope fractionation in the boundary sediments is really that. Based on Figure 1 it looks like there is a consistent positive spike in the boundary sediments but maybe not outside of uncertainty. The fact that the boundary sediments have Zn isotope ratios that are similarly heavy (+0.4-0.5permil) to some weathered sediments in the literature is surely an important consideration when interpreting the data. Instead, this is conveniently ignored.

We regret that the reviewer received the data in the original manuscript poorly. We wish to make clear to the editorial board that we did not intentionally ignore any data, and in fact the original Fig 3 plotted the published data to the best of our knowledge. The new data we were able to obtain clearly shows this unique signature that reviewer #2 alludes too. This was accomplished by obtaining samples from a depositional environment that has low authigenic Zn. These added data along with the specific edits to Figure 1 show the errors, the higher values and relative positioning of the samples for the study. Please see new Figure 1 below:

Writing: In general the writing is poor, there is an alarming lack of references, its incredibly vague in places. No data is provided to back any statements up.

We believe that this comment is likely borne out of the manuscript's history, as it was moved over from another Nature platform and had a shorter format. Where relevant, we have augmented the revised manuscript in accordance with the Reviewer's wishes by adding more references, along with stating exact values in the document. Furthermore, we have added a data table and further figure images to the main document.

Inconsistencies: The authors do not explain why the type-1 spherules have relatively unfractionated $\delta^{66}\text{Zn}$ values of +0.4 whereas they assume the impact derived material has a $\delta^{66}\text{Zn}$ value of +1.4 in the mixing calculation. The authors assume that vapor condensates should be isotopically heavy (citing Toutain et al., 2010) but that is not true. In fact Toutain et al., (2010) said the opposite.

Editorial Note: We could not find a relevant publication by Toutain et al. in 2010, and thus assume the Reviewer is referring to Toutain et al. (2008) (doi: 10.1016/j.chemgeo.2008.04.007)

We believe that this confusion has arisen from the fact that In Toutain et al. (2008), they suggest a kinetic solid-vapour fractionation factor of 1.00027 (up to 1.00045), which uses the reverse reference frame to that in most studies (e.g. on tektites), where the fractionation factor is vapour-liquid. Further adding to this confusion, the data in Toutain et al. (2008) seems to be mixed up, as in Table 4 and Figure 3, it would appear that the gas condensates are isotopically lighter in Zn. However, when dealt with in detail in the text (Section 6.3.1), they explicitly state that "As seen in Fig. 3, $\delta^{66}\text{Zn}$ of the condensates is systematically heavier than $\delta^{66}\text{Zn}$ of the gas phase at all temperatures", which they ascribe to the main condensing phase being ZnS with a stronger bond that will preferentially bind heavy Zn isotopes (as opposed to the weaker ZnCl₂ bond in the gase phase). Furthermore, their solid-vapour fractionation factor of 1.00027 (to 1.00045) is in line with this interpretation, as a fractionation factor above unity (a value of 1) indicates that the A phase is enriched in the heavier isotope relative to the B phase; when discussing α_{A-B} , where in this case the A phase is the solid as written in Toutain et al. (2008). Given the extensive discussion of this fractionation factor, and the associated solid-vapour isotope fractionation of Zn, we have chosen to assume this is what the authors intended regardless of data representation in Table 4 and Figure 3. As additional reference, to find the value for the fractionation factor of the identical reverse process, one takes the inverse of the fractionation factor. The inverse of 1.00027 is .99973, surprisingly close to that inferred in the current study, though we note Toutain et al. (2008) the second phase is solid (not liquid).

Figures: They do not add anything useful to the study. Figure 1 doesn't show the actual data, just places a best fit line through it. No uncertainties are shown. Figure 3 is supposed to show some kind of mixing line but that appears to have been left out.

We edited figure 1 above, we changed Figure 3 to show the mixing trend and then plot the modeled values.

Overall Significance: Even if a volatile depleted signature could be identified at the K-Pg boundary what

else can Zn isotopes provide about the processes that occurred or the composition of the impactor or target material. Based on this, not much.

To further strengthen the argument, the new data along with the organization of the document shows that:

1) Zn isotope signatures reflect the evaporation event. This is clearly fleshed out in the new section we added that was requested by reviewer #1, please read. The added data table to the document also shows these values upfront rather than supplemental material which this reviewer seemed to miss?

2) The model is clarified with Figure 3 that was edited along with the discussion about where the lighter Zn can be found in the boundary.

We also wish to note that the crux of the current work is to fingerprint the main operative process responsible for the K-Pg boundary, which we believe has been very robustly shown. We do not attempt—or claim—to do anything else, and frame the entire manuscript around this so as to keep it focused and coherent. This is because Zn isotopes are an incredibly robust indicator of volatilization, and are thus perhaps the best tool for this specific job, which we say. To use the data in an attempt to parse out ever finer details would be academically irresponsible due to the complicated P-T history of the vapour plume and condensation/evaporation processes operative both therein and in the ejecta and other fallout. We know this from much literature on the topic, many of which the Reviewer has in fact cited.

Overall, it's a nice set of data that is presented poorly, not well explained and the author needs to read and cite more of the relevant literature. Its not good enough for most journals, including Nature Communications.

Page by page comments:

General – it would be great to have line numbers here. For both the reviewer and for the authors trying to figure out what I am commenting on. I have definitely forgotten to provide them before, but the journal usually enforces it.

Added line numbers

Page 3:

Discern clear geochemical signatures of what?

The fact that the K-Pg sediments contain high Ir and anomalous isotopic compositions (i.e. Cr54) are pretty good indicators of an extraterrestrial impact, rather than just a simple indicator of change. I think I understand what is being said but perhaps it needs to be worded differently. The idea is, I guess, that we have no information from the K-Pg sediments about the physical processes associated with such an impact.

Yes, this is what is stated. We have edited this section to further clarify our intentions:

In the current work, the Zn isotope composition of K-Pg sedimentary rock layers have been characterized for 5 separate locations, from proximal (up to 1000 km; Mississippi and Missouri sites) to intermediate distances from (1000-5000 km; Montana and New Jersey) the impact site at Chixculub crater in the Yucatan Peninsula. All stratigraphic sequences considered contain the

characteristic spherule layer and ash associated with the impact^{12,34} (Figure 1). Samples from these sites originate from well-studied outcrops and drill cores, and the depositional environment for samples range from terrestrial (Montana), transitional (Missouri), and shallow marine (Mississippi and OPD site 165 in the Caribbean) to marine (New Jersey). The process of Zn volatilization and preservation of the unique isotopic signatures within the boundary sediments are clearly elucidated from the data, and Zn concentration data of the sediments is further leveraged to show that mixing processes occurred concomitant with post-impact fallout resulting in deposition of partially volatilized materials within K-Pg sediment boundary layers.

Page 4:

There needs to be a better introduction as to why metal stable isotopes would be important. Who cares if no data currently exists? Its not a useful lead off statement. You should introduce the moderately volatile elements and why they can provide useful information about physical processes that would, in turn, be relevant to interpreting the K-Pg sediments.

Deleted transition statement so that the paragraph solely focuses on the labile nature of Zn.

Although igneous rocks have a tight range of Zn isotope ratios there is evidence that weathering and weathered sediments can have more diverse isotopic compositions. E.g. Lv et al., 2016; Little et al., 2019...). Heterogeneity in the Zn isotope ratio can be imparted by weathering, which could be important when interpreting the K-Pg sed data.

Agreed that these statements were not ideal, we added the following two paragraphs as stated in review #1 and restated here:

Extensive study of Zn isotope compositions in Earth materials over the last 20 years indicate that the vast majority of terrestrial geologic processes do not significantly fractionate Zn isotopes (with the exception of evaporation). Redox and other kinetic reactions associated biogeochemical interactions in soils and shallow depositional environments do not produce Zn isotope values in this range^{13,14,23,24}. For instance, measured Zn isotope fractionation among geochemical products and reactants rarely approach or exceed 0.3 per mil in ore minerals and clays^{15,25-27}. While certain mineral and gas phases, and biologically-derived materials, may possess Zn isotope compositions outside the range reported above (e.g. $\delta^{66}\text{Zn}$ values ranging from -1 to +2 ‰), bulk rock values containing such phases typically do not deviate from bulk silicate Earth (BSE) and continental crustal values, because they are often only present at trace levels at the macro-/regional scale. This is most relevant for organic carbonates that can have slight enrichments in heavy Zn isotopes²⁸, however these typically contain only ~1 ppm Zn²⁹, while isotopically lighter (lower $\delta^{66}\text{Zn}$) margin and basin marine sediments (e.g. $\delta^{66}\text{Zn}$ ~0.12 ‰) can contain 10s to 100s of ppm Zn²⁴.

The stratigraphic changes in Zn isotopic signature of the boundary sediments also nullifies the potential for terrestrial alteration as a cause for the higher values at the boundary. Significantly, adjacent layers have Zn isotope compositions that are not inversely correlated to that seen within the boundary layer. This serves both to nullify leaching of heavy Zn isotopes from under or overlying rock layers as a source of $\delta^{66}\text{Zn}$ enrichment within the layer, and to highlight the otherwise typical nature of the stratigraphic sequence with regards to Zn isotope composition,

ruling out the invocation of terrestrial processes. Again, several studies corroborate the notion that even when secondary processes are present in this context, they typically do not fractionate Zn isotopes in predictable patterns or to any large degree^{15,25,30}.

Final paragraph: Zn isotopes are sensitive to volatility driven processes but are not simply controlled by them. Other processes such as mineral-melt partitioning in magmatic systems (e.g. Sossi et al., 2018) or weathering (Little et al., 2019) can fractionate Zn.

Agreed, added these references above and show that they do not fractionate Zn at this level present in the whole rock analyzed here.

An equally important aspect that could fractionate Zn relates to the generation of the melt. Multiple studies of melted mantle rocks and their residues reveal Zn does not routinely fractionate greater than 0.2 per mil^{3,19-22}.

Tektites have been analysed by several studies not referenced here: Moynier et al., 2009; Rodovska et al., 2017; Wimpenny et al., 2019...there are probably more as well!

Added these two references, Wimpenny 2019 is in the original contribution.

Page 6:

The instrument is a MC-ICP-MS and the acronym should be defined.

Added this language.

Second paragraph: This section is poorly written and should be extensively edited. Rather than referencing another paper please explain the methods to obtain Zn concentrations fully either here or in a supplemental section. In fact, it might be worthwhile moving some of isotope ratio methods to a supplemental as well.

Moved to the supplemental data and added more details about the procedures.

Approximately 200 to 300 mg of sample was dissolved in a two-stage process. First, a 6 mL mixture of ultrapure 10M HF + 8M HNO₃ was heated in 60 mL polyfluoralkyl (PFA) beaker and the solutions were dried. A second mixture of ultrapure aqua regia was used and complete digestion of the samples was visually confirmed. The Zn in post dried salts was purified using anion exchange resin (BioRad MP-1, 200-400 grain size, HCl form) under the protocol defined by³⁵; the purification protocol was done twice to ensure complete removal of matrix elements. The samples were measured by multi-collector inductively coupled plasma mass spectrometry (MC-ICP-MS) on a Thermo Neptune Plus at Rutgers University and Pennsylvania State University. Run conditions are identical to those reported in³⁶ with the instrument in low resolution mode. Sample solutions of approximately 200 ppb Zn were doped with 100 ppb Cu (NIST 976) to correct for mass bias with the exponential law³⁵. The samples were further corrected with standard-sample-standard bracketing using the AA-ETH Zn isotope³⁷. Data have been reported relative to JCM Lyon standard (which is no longer available) by applying a +0.27‰ difference³⁸⁻⁴⁰. The error of the standard compared to itself throughout the 4 session is ±0.05‰ and is considered a conservative estimation of error as duplicate analyses of the same solutions and complete procedural duplicates of the USGS BVHO-2 rock standard (+0.01 ± 0.03‰, 2σ, n=8, overlapping values found in⁴¹) are significantly less.

Zn concentration measurements were conducted via quadrupole ICP-MS, and calculated gravimetrically using 100 mg aliquots of the same powdered samples. The ICP-MS instrument conditions and data reduction protocols are found in⁴². BVHO-2 Zn concentration were within 8% of reported values. Scanning electron microscope (SEM) images were obtained on a JEOL 6460 SEM at Juniata College.

Page 7:

What is significant fractionation of Zn? This means different things to different people.

What does falling in this category mean?

Why would these layers be inversely correlated?

Is there any indication that leaching of heavy isotopes from other layers would occur? Reference a study that shows this here.

This section was completely changed as described above and see changes requested by reviewer #1 which point to the same issues. We note that in the original manuscript we indeed discussed potential leaching from adjacent layers, and negated this possibility by showing that there is no inverse Zn isotope excursion in adjacent strata – this would generally be expected by mass balance, unless the leachate was completely removed, and there is no evidence for this.

Replace too with to

Deleted the sentence

There needs to be some quantification of Zn fractionation in weathered continental sediments for context. Provide numbers and references. I have seen data that shows $\delta^{66}\text{Zn}$ values of >0.5 permil in weathered sediments (Yv et al., 2016 etc...) This first paragraph is far too vague.

We have provided clear context with the values now stated:

Extensive study of Zn isotope compositions in Earth materials over the last 20 years indicate that the vast majority of terrestrial geologic processes do not significantly fractionate Zn isotopes (with the exception of evaporation). Redox and other kinetic reactions associated biogeochemical interactions in soils and shallow depositional environments do not produce Zn isotope values in this range^{13,14,23,24}. For instance, measured Zn isotope fractionation among geochemical products and reactants rarely approach or exceed 0.3 per mil in ore minerals and clays^{15,25-27}. While certain mineral and gas phases, and biologically-derived materials, may possess Zn isotope compositions outside the range reported above (e.g. $\delta^{66}\text{Zn}$ values ranging from -1 to +2 ‰), bulk rock values containing such phases typically do not deviate from bulk silicate Earth (BSE) and continental crustal values, because they are often only present at trace levels at the macro-/regional scale. This is most relevant for organic carbonates that can have slight enrichments in heavy Zn isotopes²⁸, however these typically contain only ~1 ppm Zn²⁹, while isotopically lighter (lower $\delta^{66}\text{Zn}$) margin and basin marine sediments (e.g. $\delta^{66}\text{Zn}$ ~0.12 ‰) can contain 10s to 100s of ppm Zn²⁴.

The stratigraphic changes in Zn isotopic signature of the boundary sediments also nullifies the potential for terrestrial alteration as a cause for the higher values at the boundary. Significantly, adjacent layers have Zn isotope compositions that are not inversely correlated to that seen within the boundary layer. This serves both to nullify leaching of heavy Zn isotopes from under or overlying rock layers as a source of $\delta^{66}\text{Zn}$ enrichment within the layer, and to highlight the otherwise typical nature of the stratigraphic sequence with regards to Zn isotope composition,

ruling out the invocation of terrestrial processes. Again, several studies corroborate the notion that even when secondary processes are present in this context, they typically do not fractionate Zn isotopes in predictable patterns or to any large degree^{15,25,30}.

Why don't you start the second paragraph by saying that the K-Pg sediments have higher $\delta^{66}\text{Zn}$ values than strata above and below?

This is changed both in the new paragraphs and new section title to explain significance.

What is the significant uptick? Provide numbers here. This is far too vague. Figure 1 is not very clear – is the uptick outside of uncertainty of the technique?

Yes, in the original manuscript we noted that at numerous sites the excursion is statistically significant, and where it borders on otherwise, we also noted. To alleviate concerns and make this more explicit, we have added errors bars to Figure 1 as stated and demonstrated above.

Again, paragraph 2 suffers from being too vague. I don't know that significance of the changes in Zn isotope ratio from the text or the figure so its hard to evaluate the following interpretation.

Paragraph 3: Do you mean the depositional environments in this study or in general? Where do these numbers come from?

Stated above and added figures for comparison in the text.

Why should the Zn isotope ratio at the boundary vary as a consequence if fractionation is imparted by evaporative loss? Unless you are talking about the addition of fractionated material from the debris cloud (i.e. two-component mixing).

We regret that it was not apparent that this is exactly what we are talking about. Throughout the manuscript, we have clarified text so that this is more explicit and readily apparent. The key point is that the K-Pg boundary sediment layer hosts material—e.g. spherules—with a high $\delta^{66}\text{Zn}$ caused by the partial evaporation of their Zn contents during drag heating along their ballistic trajectories from the impact site to their final sedimentary location.

Page 8:

Figure 2 doesn't show typical crustal compositions so we can't compare the data. Unless CC represents the crust? But that acronym is typically used to denote carbonaceous chondrites...

We define CC as continental crust in the figure caption.

Paragraph 2: Tektites are not sediments from the K-Pg boundary. Sediments from the K-Pg boundary have high Ir and Cr isotope anomalies that derive from the impactor (as stated in the introduction). As far as I know these have not been found in tektites previously. There must be estimates for the amount of impactor material required to generate these anomalies in the K-Pg sediments. Provide them here. I don't think that you have proven that the Zn fractionation could not be a signal from the impactor without providing a mass balance calculation based on the amount of impactor material in the K-Pg sediments.

Agreed, we added the paragraph and section with mass balance to explain this weakness in the paper.

Paragraph 3: You have already mentioned tektites so why are you defining them halfway through the manuscript?

Deleted.

The second sentence is poorly written. What you mean is 'Samples derived from impact events such as...are relatively depleted in Zn and have relatively heavy $\delta^{66}\text{Zn}$ values, indicating loss of Zn by volatilization.

Reworded this sentence.

Page 9:

The mixing approach is sensible but Figure 3 does not help the discussion. It doesn't illustrate mixing at all. Where are the mixing lines that are described?!

Fixed Figure 3 to just show the mixing relationship and the modelling of this information. This figure has two panels and shows the different mixing trends explained in the text and how the data fit modeled distillation curves.

How does this compare with other estimates of impactor material in the K-Pg sediments?

Added references to other geochemical articles dealing with this issue, such as:

43-45

Paragraph 3: A great deal of the tektite references are missing here.

The data in S2 do not comply with a simple Rayleigh distillation curve. They are highly scattered and the data appear to sit on different curves. The assertion that these data are better than the available tektite data is completely unfounded and not very scientific. Moynier's data appears more convincing based on Figure S2. Please also see Wimpenny et al., (2019) who also provide a well-defined Rayleigh fractionation relationship in tektites.

We agree that there is no scientific merit in claiming supremacy in relative goodness of fit between our data and that in Moynier et al. (2009), and we have removed the associated parenthetical. Our goal was simply to point out that—especially in the terrestrial and transitional environments, where there is less signal suppression—that the data fit very well within the reported range of empirical fractionation factors (i.e. 0.9999 to 0.9998), and are less dispersed than the high $\delta^{66}\text{Zn}$ data of Moynier et al. (2009)

Wimpenny et al. (2019) is largely focused on experimental work. The novel Australite data in their work only encompass samples that have presumably already lost significant Zn, and as such they combine their data with that of Moynier et al. (2009) in order to determine an range of fractionation factors that envelope the data. This envelope is ranges from 0.998 to 0.9997, or roughly an order of magnitude larger than that used in our modelling. Thus, we find it difficult to claim this as well-defined Rayleigh distillation, relative to our own work or not.

Lastly on this, point, in the original and revised manuscript, no claim is ever made that what is observed is simple Rayleigh distillation, and in fact at multiple junctures we have made a point to explicitly point out that these are approximated, empirical constraints, where the purpose is to show that by modelling

“attenuated” isotope fractionation due to signal dilution adequately envelopes the data from a scientific standpoint.

Page 10:

Paragraph 1: This is a highly speculative interpretation of the alpha values given the scatter in the data. The problem with interpreting tektite data is that the precursor material is not well constrained, particularly for australites which form the majority of current sample sets. This makes the alpha for tektite data relatively imprecise. This should be considered because currently I believe the data are being over interpreted. The best you could say is there may be some similarity in alpha during evaporative loss if ~10% of the K-Pg material derived from volatile depleted impact debris. Its consistent rather than incredibly concordant.

We have removed the word “incredibly” as this has no scientific metric attached to it. We are in effect saying that there is similarity between the enveloping alpha values of our dataset and that seen in work on tektites. We stand by our wording otherwise, and do not believe we are over-interpreting the data given its consistency with evaporative processes (e.g. in tektites and Sudbury data), and given that at every relevant point we clearly state where interpretations are tentative, speculative or the like.

Where is the Sudbury data from?

Kamber and Schoenberg 2020

I don't think you can say much about the process other than the sediments consist of material that has lost volatiles.

Yes, this is exactly what we are saying, and we are very glad to hear that the reviewer agrees. To date, no one has identified a marker of an evaporation process at the boundary, leaving causal relations nebulous. We have identified a bolide impact as the culprit at the K-Pg using novel Zn isotope data as a fingerprinting tool, as it is very well suited for fingerprinting volatilization. We strongly believe that this is of scientific merit and value to the community, regardless of how speculative the finer details are.

Paragraph 2: This repeats much of what was stated on page 8. If the $\delta^{66}\text{Zn}$ values are within uncertainty of the continental crust can you really state that the marine sites have elevated $\delta^{66}\text{Zn}$ values?

Yes, we can with the new data as stated above.

Paragraph 3: Wouldn't impact melt spherules also be the source of heavy Zn in the terrestrial sections as well?

Based on the information in the supplemental the spherules have $\delta^{66}\text{Zn}$ values of +0.39 and +0.43... If the impact spherules are the source of isotopically heavy Zn isn't there a mass balance issue? The spherules have $\delta^{66}\text{Zn}$ values that are similar to UCC. Therefore, to shift the $\delta^{66}\text{Zn}$ value of the boundary layer would require the majority of Zn to be derived from the impact debris, rather than the 10% that is assumed on page 9.

We regret to see that the reviewer may have misinterpreted our interpretation of the dataset, especially as regards the very limited number of Zn isotope analyses from melt spherules. There are ashes and other types of materials that could possess this higher signature, and we fully recognize that only two

analyses of billions of spherules that exist on Earth do not fully characterize all spherules and/or all types of impact related ejecta. We added the following language to acknowledge this.

The likely source of the higher Zn isotope values found in the marine samples is related to the presence of type 1 impact spherules and ash (splash-form melt droplets). The type 1 spherules that yielded elevated Zn isotope values (+0.41‰ and +0.44‰) and elevated Zn concentrations. The melt induced droplets have nearly identical Zn isotope compositions to the melt sheet and fallout crater fill found in Sudbury at +0.45±0.07‰ (mean calculated by elimination of high and low outliers). During impact melt generation, Zn has been observed to concentrate in melt droplets⁴⁶. Thus, these concentrated spherule

Page 11:

I assume that this Zn data from Sudbury is from Kamber & Schoenberg? Why is this not cited?

It was cited 3 times in the original document. It has now been cited 4 times in the revised document.

But the isotopic value of impact spherules is not elevated compared to sediments from the terrestrial boundary layers...Some of the marine sediments have $\delta^{66}\text{Zn}$ values that are very similar to the impact spherules which would imply that the spherules contribute >90% of Zn to the rock based on the arguments used here. Also, the assumed composition of the impact spherules in the mixing calculation was +1.4 permil. Why do the spherules that were analysed have $\delta^{66}\text{Zn}$ values that are so much lower? There is a lack of consistency here meaning that either I am missing something (which is possible) or the explanation needs to be rethought.

We added more details around this idea of the spherules as described in response to a similar issue in reviewer #1.

Where does this 100-200ppm estimate come from? Is it from analyses of CM chondrites (e.g. Luck et al., 2005)? What if the organics in the limestone were preferentially volatilized? Could they have contributed disproportionately to the boundary layer?

Paragraph 3: Hasn't this been explained already?

Similar observations have also been made for Ga (Wimpenny et al., 2020)

Page 12:

All estimates of volatilization during impact at Chixculub are well over 1000degC, at which point BOTH organic and carbonate carbon are volatilized

We're not discussing Ga here, nor are we invoking unfounded possibilities that suffer from infinite regress ("but what if", ah okay, "but what if")

Paragraph 1 – The key problem is pressure into which evaporation takes place. The theoretical kinetic fractionation factor can only be attained in a vacuum. Read a few papers by Richter for more information on this. Higher vapor pressures suppress the fractionation thus the alpha from volatile loss in terrestrial systems will be closer to 1. There is plenty of data out there and references that could be made. For example, see Yu et al., (2003) who show a relationship between K isotope fractionation and vapor pressure. See associated work in Wang & Jacobsen 2016. Unfortunately, this paragraph makes use of none of the prior work and as a result is not very informative. It reads like the suppression of alpha is a new discovery. It's not.

Have now added further discourse in revised manuscript in Binary Mixing section

Paragraph 2: Gas condensates on the Moon are highly enriched in light Zn, as one would expect in a kinetic process, where the forward reaction favors the lighter isotope. Fumeroles are a more complicated system, where the composition of the gas phase also reflects the evolving reservoir. However, if you actually read Toutain et al., (2010) you will see that the fumarolic gases are interpreted to condense light Zn, but the remaining Zn in the gas becomes heavier as the gas continues to ascend.

Thus, the first condensates in the vent would be light and then get progressively heavier. Any gas condensate would thus be enriched in light Zn isotopes contrary to the assertion here for type-2 spherules.

We have rephrased this sentence in the revised manuscript to be more clear, and would like to reference our previous response regarding the work of Toutain et al. (2010) for further discourse regarding Zn isotope systematics during condensation.

Figure 1 – Add the data points and uncertainties not just a best fit line. This figure is useless right now as I have no idea whether the ‘excursions’ are real or within uncertainty of the other data.

Figure 1 edited and added error bars so that sample locations in the cores are seen.

Figure 2 – What is CC? What is a gas cond and where are the data from? What tektite data is used here?

Added figure caption and tied references with gas condensate and CC in the text.

Figure 3 – I do not understand how this demonstrates a mixing relationship.

Changed figure and added clear demonstration of the mixing trend.

Reviewer #3 (Remarks to the Author):

This is a well-argued and well-written manuscript, with interesting and obviously high-quality data. I have no concerns about the general idea, the data, or the interpretation and conclusions. Most of the questions I asked myself when reading the manuscript were answered somewhere in the manuscript. The authors could strengthen their arguments by maybe rewriting or amplifying some of their points, as follows.

- In the abstract, there are some what I'd say filling sentences or sentence parts. It would be more succinct to clearly point out what the goals of the present work were/are (what is the scientific question that the authors set out to answer); explain what variety of materials was analyzed and why, explain why alteration etc does not affect the Zn isotopic composition, and clearly state even in the abstract that volcanism cannot explain the observed values.

Agreed and added this sentence to the abstract:

Neither volcanic events nor secondary alteration during weathering and diagenesis explain the Zn isotope signatures present.

- In general I also thought that the question of (Deccan) volcanism as a possible source - given how a small but vocal minority harps on this topic - should be more clearly dealt with, and in the text more clearly stated why the data do not agree with such an idea.

To clarify this issue, we adjusted the subtitles to each section, so that we show the implications of the data and edited the last section to more convincingly show volcanism cannot cause this signature.

- The connection to the processes that have been observed in tektites could also be made more clearly; the authors note that in some cases K-Pg spherules were analyzed, but were those glassy or not? Some rudimentary petrographic descriptions (a couple of sentences) of the samples would be useful in the main text.

Agreed, we have modified the discussion as requested by reviewer #2 in the earlier part of the response.

- I am a bit sad that no data tables are present in the main text. These days too much important material gets pushed to supplements.... if space allows it it would be good to put a table there.

Agreed, we added the data table and moved the modeling figures into the document and moved the methods to the supplemental data.

Otherwise the figures look ok, the references appear in order and up to date, so I have no additional comments there.

I look forward to seeing this interesting work published.

Christian Koeberl
University of Vienna, Austria

(Jan. 2, 2021)

- 1 Kamber, B. S. & Schoenberg, R. Evaporative loss of moderately volatile metals from the
superheated 1849 Ma Sudbury impact melt sheet inferred from stable Zn isotopes. *Earth and
Planetary Science Letters* **544**, 116356, doi:<https://doi.org/10.1016/j.epsl.2020.116356> (2020).
- 2 Lodders, K. & Fegley, B. *The planetary scientist's companion*. (Oxford University Press on
Demand, 1998).
- 3 Sossi, P. A., Nebel, O., O'Neill, H. S. C. & Moynier, F. Zinc isotope composition of the Earth and its
behaviour during planetary accretion. *Chemical Geology* **477**, 73-84 (2018).
- 4 Mahan, B., Moynier, F., Beck, P., Pringle, E. A. & Siebert, J. A history of violence: Insights into
post-accretionary heating in carbonaceous chondrites from volatile element abundances, Zn
isotopes and water contents. *Geochimica et Cosmochimica Acta* **220**, 19-35,
doi:<https://doi.org/10.1016/j.gca.2017.09.027> (2018).
- 5 Chen, H., Savage, P. S., Teng, F.-Z., Helz, R. T. & Moynier, F. Zinc isotope fractionation during
magmatic differentiation and the isotopic composition of the bulk Earth. *Earth and Planetary
Science Letters* **369-370**, 34-42, doi:<https://doi.org/10.1016/j.epsl.2013.02.037> (2013).
- 6 Moynier, F. *et al.* Nature of volatile depletion and genetic relationships in enstatite chondrites
and aubrites inferred from Zn isotopes. *Geochimica et Cosmochimica Acta* **75**, 297-307 (2011).
- 7 Pringle, E. A., Moynier, F., Beck, P., Paniello, R. & Hezel, D. C. The origin of volatile element
depletion in early solar system material: Clues from Zn isotopes in chondrules. *Earth and
Planetary Science Letters* **468**, 62-71, doi:<https://doi.org/10.1016/j.epsl.2017.04.002> (2017).
- 8 Paniello, R. C., Day, J. M. & Moynier, F. Zinc isotopic evidence for the origin of the Moon. *Nature*
490, 376-379 (2012).
- 9 Kato, C., Moynier, F., Valdes, M. C., Dhaliwal, J. K. & Day, J. M. Extensive volatile loss during
formation and differentiation of the Moon. *Nature communications* **6**, 1-4 (2015).
- 10 Rodovská, Z. *et al.* Implications for behavior of volatile elements during impacts—Zinc and
copper systematics in sediments from the Ries impact structure and central European tektites.
Meteoritics & Planetary Science **52**, 2178-2192 (2017).
- 11 Kamber, B. S. & Petrus, J. A. The influence of large bolide impacts on Earth's carbon cycle.
Elements: An International Magazine of Mineralogy, Geochemistry, and Petrology **15**, 313-318
(2019).
- 12 Schulte, P. *et al.* The Chicxulub asteroid impact and mass extinction at the Cretaceous-Paleogene
boundary. *Science* **327**, 1214-1218 (2010).
- 13 Lv, Y., Liu, S.-A., Zhu, J.-M. & Li, S. Copper and zinc isotope fractionation during deposition and
weathering of highly metalliferous black shales in central China. *Chemical Geology* **445**, 24-35
(2016).
- 14 Little, S. H. *et al.* Cu and Zn isotope fractionation during extreme chemical weathering.
Geochimica et Cosmochimica Acta **263**, 85-107, doi:<https://doi.org/10.1016/j.gca.2019.07.057>
(2019).

- 15 Mondillo, N., Wilkinson, J. J., Boni, M., Weiss, D. J. & Mathur, R. A global assessment of Zn isotope fractionation in secondary Zn minerals from sulfide and non-sulfide ore deposits and model for fractionation control. *Chemical Geology* **500**, 182-193, doi:<https://doi.org/10.1016/j.chemgeo.2018.09.033> (2018).
- 16 Mason, T. F. *et al.* Zn and Cu isotopic variability in the Alexandrinka volcanic-hosted massive sulphide (VHMS) ore deposit, Urals, Russia. *Chemical Geology* **221**, 170-187 (2005).
- 17 Moynier, F. *et al.* Isotopic fractionation of zinc in tektites. *Earth and Planetary Science Letters* **277**, 482-489 (2009).
- 18 Wimpenny, J. *et al.* Experimental determination of Zn isotope fractionation during evaporative loss at extreme temperatures. *Geochimica et Cosmochimica Acta* **259**, 391-411, doi:<https://doi.org/10.1016/j.gca.2019.06.016> (2019).
- 19 Doucet, L. S., Mattielli, N., Ionov, D. A., Debouge, W. & Golovin, A. V. Zn isotopic heterogeneity in the mantle: A melting control? *Earth and Planetary Science Letters* **451**, 232-240 (2016).
- 20 Doucet, L. S., Laurent, O., Mattielli, N. & Debouge, W. Zn isotope heterogeneity in the continental lithosphere: New evidence from Archean granitoids of the northern Kaapvaal craton, South Africa. *Chemical Geology* **476**, 260-271 (2018).
- 21 Zhang, G. *et al.* Zinc isotopic composition of the lower continental crust estimated from lower crustal xenoliths and granulite terrains. *Geochimica et Cosmochimica Acta* **276**, 92-108, doi:<https://doi.org/10.1016/j.gca.2020.02.030> (2020).
- 22 McCoy-West, A. J., Fitton, J. G., Pons, M.-L., Inglis, E. C. & Williams, H. M. The Fe and Zn isotope composition of deep mantle source regions: Insights from Baffin Island picrites. *Geochimica et Cosmochimica Acta* **238**, 542-562 (2018).
- 23 Little, S. H., Vance, D., Walker-Brown, C. & Landing, W. M. The oceanic mass balance of copper and zinc isotopes, investigated by analysis of their inputs, and outputs to ferromanganese oxide sediments. *Geochimica et Cosmochimica Acta* **125**, 673-693, doi:<http://dx.doi.org/10.1016/j.gca.2013.07.046> (2014).
- 24 Little, S. H., Vance, D., McManus, J. & Severmann, S. Key role of continental margin sediments in the oceanic mass balance of Zn and Zn isotopes. *Geology* **44**, 207-210 (2016).
- 25 Borrok, D. M., Nimick, D. A., Wanty, R. B. & Ridley, W. I. Isotopic variations of dissolved copper and zinc in stream waters affected by historical mining. *Geochimica et Cosmochimica Acta* **72**, 329-344 (2008).
- 26 Bigalke, M., Weyer, S., Kobza, J. & Wilcke, W. Stable Cu and Zn isotope ratios as tracers of sources and transport of Cu and Zn in contaminated soil. *Geochimica et Cosmochimica Acta* **74**, 6801-6813 (2010).
- 27 Guinoiseau, D. *et al.* Zinc and copper behaviour at the soil-river interface: New insights by Zn and Cu isotopes in the organic-rich Rio Negro basin. *Geochimica et Cosmochimica Acta* **213**, 178-197, doi:<https://doi.org/10.1016/j.gca.2017.06.030> (2017).
- 28 Pichat, S., Douchet, C. & Albarede, F. Zinc isotope variations in deep-sea carbonates from the eastern Equatorial Pacific over the last 175 ka. *Earth and Planetary Science Letters* **210**, 167-178 (2003).
- 29 Hartree, R. & Veizer, J. Lead and zinc distribution in carbonate rocks. *Chemical Geology* **37**, 351-365, doi:[https://doi.org/10.1016/0009-2541\(82\)90088-2](https://doi.org/10.1016/0009-2541(82)90088-2) (1982).
- 30 Borrok, D. M. *et al.* Separation of copper, iron, and zinc from complex aqueous solutions for isotopic measurement. *Chemical Geology* **242**, 400-414 (2007).
- 31 Sossi, P. A. *et al.* An experimentally-determined general formalism for evaporation and isotope fractionation of Cu and Zn from silicate melts between 1300 and 1500 °C and 1 bar. *Geochimica et Cosmochimica Acta* **288**, 316-340, doi:<https://doi.org/10.1016/j.gca.2020.08.011> (2020).

- 32 Liu, S.-A. *et al.* Zinc isotope evidence for intensive magmatism immediately before the end-Permian mass extinction. *Geology* **45**, 343-346 (2017).
- 33 Ravizza, G. & Peucker-Ehrenbrink, B. Chemostratigraphic evidence of Deccan volcanism from the marine osmium isotope record. *Science* **302**, 1392-1395 (2003).
- 34 Artemieva, N. & Morgan, J. Modeling the formation of the K–Pg boundary layer. *Icarus* **201**, 768-780 (2009).
- 35 Marechal, C. N., Telouk, P. & Albarede, F. Precise analysis of copper and zinc isotopic compositions by plasma-source mass spectrometry. *Chemical Geology* **156**, 251-273 (1999).
- 36 Matt, P., Powell, W., Mathur, R. & deLorraine, W. F. Zn-isotopic evidence for fluid-assisted ore remobilization at the Balmat Zinc Mine, NY. *Ore Geology Reviews* **116**, 103227, doi:<https://doi.org/10.1016/j.oregeorev.2019.103227> (2020).
- 37 Archer, C. *et al.* Inter-calibration of a proposed new primary reference standard AA-ETH Zn for zinc isotopic analysis. *Journal of Analytical Atomic Spectrometry* **32**, 415-419, doi:10.1039/C6JA00282J (2017).
- 38 Albarede, F. *et al.* Precise and accurate isotopic measurements using multiple-collector ICPMS. *Geochimica et Cosmochimica Acta* **68**, 2725-2744 (2004).
- 39 Albarede, F. & Beard, B. Analytical Methods for Non-Traditional Isotopes. *Reviews in Mineralogy and Geochemistry* **55**, 113-152 (2004).
- 40 Albarede, F. The Stable Isotope Geochemistry of Copper and Zinc. *Reviews in Mineralogy and Geochemistry* **55**, 409-427 (2004).
- 41 Chapman, J. B., Mason, T. F. D., Weiss, D. J., Coles, B. J. & Wilkinson, J. J. Chemical separation and isotopic variations of Cu and Zn from five geological reference materials. *Geostandards and Geoanalytical Research* **30**, 5-16 (2006).
- 42 Yesavage, T. *et al.* Fe cycling in the Shale Hills Critical Zone Observatory, Pennsylvania: An analysis of biogeochemical weathering and Fe isotope fractionation. *Geochimica et Cosmochimica Acta* **99**, 18-38, doi:<https://doi.org/10.1016/j.gca.2012.09.029> (2012).
- 43 Gelinas, A. *et al.* Osmium isotope constraints on the proportion of bolide component in Chicxulub impact melt rocks. *Meteoritics & Planetary Science* **39**, 1003-1008, doi:<https://doi.org/10.1111/j.1945-5100.2004.tb00941.x> (2004).
- 44 Tagle, R. *et al.* Platinum group elements in impactites of the ICDP Chicxulub drill core Yaxcopoil-1: Are there traces of the projectile? *Meteoritics & Planetary Science* **39**, 1009-1016, doi:<https://doi.org/10.1111/j.1945-5100.2004.tb00942.x> (2004).
- 45 Goderis, S. *et al.* Reevaluation of siderophile element abundances and ratios across the Cretaceous–Paleogene (K–Pg) boundary: Implications for the nature of the projectile. *Geochimica et Cosmochimica Acta* **120**, 417-446 (2013).
- 46 Belza, J. *et al.* High spatial resolution geochemistry and textural characteristics of ‘microtektite’ glass spherules in proximal Cretaceous–Paleogene sections: Insights into glass alteration patterns and precursor melt lithologies. *Geochimica et Cosmochimica Acta* **152**, 1-38, doi:<https://doi.org/10.1016/j.gca.2014.12.013> (2015).

REVIEWER COMMENTS

Reviewer #1 (Remarks to the Author):

As a reviewer of the original submission, I feel that the majority of my concerns/suggestions have been addressed satisfactorily. For my taste, the modelling of the revised version is a bit too strongly focused on tektites. I think the general readers of the journal would benefit from a reminder that tektites are mainly splash curtain ejecta but that their isotope systematics and volatilisation history may not be perfect analogues for the bulk of the melt sheet, which contains much more overall mass. As a partly submerged impact structure Chicxulub almost certainly experienced explosive ejection of melt sheet material upon re-entry of seawater into the crater and I'm therefore not sure that tektites necessarily represent the best analogue for the extent of isotope fractionation that might have been experienced by the ejecta material found in the analysed sedimentary rocks. I'm not asking for a different model, simply for a statement that better explains why tektites per se may not be the best proxies for expected isotope ratios in the isotopically heavy end-member.

Reviewer #2 (Remarks to the Author):

This is a revised version of a manuscript that I previously reviewed for Nature Communications. Its an interesting subject and I'm sure would be of interest to readers of this journal. The observation that boundary layers from the K-Pg boundary are depleted in Zn and have a heavier Zn isotope ratio than the country rock is consistent with input of volatile depleted material from an impact event. So I do agree with the authors about the main thrust of the manuscript. However, there are still problems with it. The main problems being the following:

- Poorly organized. I made several comments saying that paragraphs could be consolidated due to repetition. The best example being paragraphs on L304 and 316. Honestly, this paper could be ~1/4 shorter and tightening it up would greatly improve the readability. Sometimes its useful to have a fresh set of eyes when doing this. But that's OK, because there are 8 co-authors so maybe one of them could help out.
- The key take home message is that the Zn isotope ratio of the boundary layer is heavier than the country rock and that this heavy signal can only come from impact debris that is isotopically heavy. However, I don't think this is well argued right now. There needs to be clear explanation that terrestrial weathering processes (e.g. clay formation) do not cause this type of fractionation. Current discussion probably touches on this but is poorly organized. There also needs to be better discussion of why this heavy signal could not come from meteorites. For reference, enstatite chondrites have $\delta^{66}\text{Zn}$ values up to +7permil (Moynier et al., 2011) so there should be thorough discussion of why impactors with such heavy Zn isotope ratios could not be responsible.
- Figure 2 – I don't understand the point of the figure. Comparing the sediments measured here with literature data from the continental crust doesn't seem useful. You need a clear way to show that the boundary layer rocks have heavier $\delta^{66}\text{Zn}$ values than the adjacent sediment layers but this isn't it.
- Check references are correct. Make sure that all tektite data is considered in addition to Moyniers 2009 paper.

I think these issues are fairly straightforward to address although would involve some reorganization of the discussion which is non-trivial. My recommendation is that the author perform the changes under the guidance of the editor and that once complete the manuscript will be acceptable for publication.

L100- But to what degree is this heavier than rocks/sediments that have not experienced volatile loss? Can the volatile depletion signal be unambiguously identified? This is a key point regarding why you would want to use Zn isotopes for this study.

L145 - Figure 2 doesn't show this. It shows that the sediments and spherules contain heavier Zn isotope ratios than rocks from the continental crust (on average).

L147 - How do you know the composition of the bolide? Meteorites have a wide range of Zn isotope ratios, some of which are very heavy (e.g. Moyniers work on enstatite chondrites).

L152 - These points aren't on Figure 2.

L155 - This is not correct. Enstatite chondrites have Zn isotope ratios up to +7 permil heavier than the bulk

silicate earth (Moynier et al., 2011).

L159 - Only one sample is relatively heavy. The rest have $\delta^{66}\text{Zn}$ values that are found in crustal rocks that have not experienced volatilization. I think it is more important to stress the isotopic difference between the boundary layer and the country rocks. Can you resolve an isotopic difference in each profile? What is that difference?

L163 - Didn't you discount these processes in the previous paragraph?

L174 - But isn't the point here that heating (and melting) was accompanied by degassing of isotopically light Zn? Are you talking about magmatic differentiation processes (i.e. mineral-melt partitioning)?

L182 - What about basic silicate weathering processes like clay formation?

L197 - Is this process even likely to happen? Is there any evidence that leaching of isotopically heavy Zn from sediment layers can impart a heavy signature to neighbouring rocks? It seems overly complicated. Wouldn't a simpler scenario be that a period of intense weathering leached away isotopically light Zn from the boundary layer?

L203 - A problem here is that there hasn't been enough effort to unambiguously show that the boundary layers have higher Zn isotope ratios than the country rock. Figure 2 doesn't show this (despite what is claimed on L145). And Fig. 1 is too small to see easily. Is the isotopic difference between boundary layer and country rock always statistically significant? Also, there is no clear argument provided that material from the impactor itself couldn't have contributed isotopically heavy Zn to the boundary layer. These should be key parts of the discussion. The rest of the section (L144-203) is quite muddled. You start off by stating that target rocks and meteorites don't contain isotopically heavy Zn that could explain the single heaviest $\delta^{66}\text{Zn}$ value (L150-159) and then go on to repeat some of this discussion later (L176-192). This section should be consolidated and discussion of meteorites expanded to include samples such as enstatite chondrites.

L203 - I do agree that the heavy $\delta^{66}\text{Zn}$ values at the boundary layers are consistent with the addition of volatile depleted material. I just don't think the current discussion has been optimized to show this.

L208 - The key observation is relatively simple; the increase in $\delta^{66}\text{Zn}$ correlates with a decrease in Zn concentration. Is this what is seen in the K-Pg boundary sediments?

L220 - Why would fast weathering impart a light signature to the oceans? Does this mean that weathering residue is isotopically heavy? Why is volcanic ash isotopically light? What process causes this? What are the implications of this for interpretation of K-Pg boundary sediments?

L246 - The reference for the isotopic composition of Zn in tektites is incorrect. Fig. 3 references Moynier's 2009 paper but there are at least 3 studies that I know of that contain Zn isotope data from tektites, and possibly more exist. They should all be referenced in this study. I said this in my previous review.

L266 - This is because different tektites have experienced different extents of Zn loss.

L290-299 - I don't really understand the point being made here. It seems to repeat the point made on L282 about mixing of ejecta with sediment layers. I'd just delete these lines and skip to the final sentence on L299 about increasing $\delta^{66}\text{Zn}$ values corresponding to decreasing Zn concentrations.

L304-316 and L317-330 - These paragraphs repeat each other and also repeat discussion on L245-254 about contamination of the impact signal by sediment input. These should be consolidated. I honestly don't understand - did nobody read this?!

L335 - But if carbonates don't contain much Zn how can they be the source of the isotopically heavy Zn in the boundary layer? Couldn't you rule them out based on this?

L341-358 - The suppression of α was nicely explained on L265-289. It's unnecessary to repeat that here. Again, these sections should be consolidated.

L363-369 - Break this sentence up.

L372 - This is a long-winded way of saying that impacts don't contribute enough Zn mass to perturb the bulk silicate earth Zn composition. Are there estimates of total extraterrestrial material to have hit the Earth (vs total silicate mass)?

L375 - What is the time lag between the light sample at the top of the boundary and the heavy sample at the base of the boundary, which is presumably from when the impact happened? I assume that amount of sedimentation represents a decent amount of geological time (hundreds/thousands of years?) Would you expect fine particles with isotopically light Zn isotope ratios to float around the atmosphere for such a long time? This all seems like a highly speculative hypothesis. Could a more mundane explanation such as a shift in weathering/climate/ocean conditions following the impact explain a drop in the Zn isotope ratio?

Reviewer #3 (Remarks to the Author):

The revision addresses all of the points raised in review and I have no further comments, and recommend acceptance of this manuscript.

Best regards,
Christian Koeberl
(Univ. Vienna, Austria)

We appreciate the time and effort of the reviewers and we provide responses to the specific comments in *italics*, and indicate changes made to the revised manuscript using *red italics*.

Reviewer #1 (Remarks to the Author):

As a reviewer of the original submission, I feel that the majority of my concerns/suggestions have been addressed satisfactorily. For my taste, the modelling of the revised version is a bit too strongly focused on tektites. I think the general readers of the journal would benefit from a reminder that tektites are mainly splash curtain ejecta but that their isotope systematics and volatilisation history may not be perfect analogues for the bulk of the melt sheet, which contains much more overall mass. As a partly submerged impact structure Chicxulub almost certainly experienced explosive ejection of melt sheet material upon re-entry of seawater into the crater and I'm therefore not sure that tektites necessarily represent the best analogue for the extent of isotope fractionation that might have been experienced by the ejecta material found in the analyzed sedimentary rocks. I'm not asking for a different model, simply for a statement that better explains why tektites per se may not be the best proxies for expected isotope ratios in the isotopically heavy end-member.

Agreed, the rationale behind the focus on tektites has been driven by the available data at this moment and the fact these have been documented to have volatile isotopic signatures. We added the following language in the modeling section.

Line 262 added:

Modeling and discussion herein focus on tektite compositions as the high $\delta^{66}\text{Zn}$, low Zn concentration end member proxy. Other fall out material could possess this volatilized signature, unfortunately, however tektites are the most studied/representative materials available to model.

Line 273 added

'a devolatilized impact fallout material composition with lower Zn concentration (1ppm) and heavy Zn isotope signature ($\delta^{66}\text{Zn} \sim 1.8 \text{‰}$), modeled after average tektite compositions.

Throughout the R1 and R2 manuscript, the binary mixing and use of tektites as a proxy are repeatedly and explicitly noted for their simplicity and limitations, attesting to our efforts to clearly point out that we do not consider this as an apples-to-apples comparison, but rather consider it as the best proxy system available to investigate given the scientific limitations on data present in the literature and within the manuscript.

Reviewer #2 (Remarks to the Author):

This is a revised version of a manuscript that I previously reviewed for Nature Communications. It is an interesting subject and I'm sure would be of interest to readers of this journal. The observation that boundary layers from the K-Pg boundary are depleted in Zn and have a heavier Zn isotope ratio than the

country rock is consistent with input of volatile depleted material from an impact event. So I do agree with the authors about the main thrust of the manuscript. However, there are still problems with it. The main problems being the following:

- Poorly organized. I made several comments saying that paragraphs could be consolidated due to repetition. The best example being paragraphs on L304 and 316.

Honestly, this paper could be ~1/4 shorter and tightening it up would greatly improve the readability. Sometimes its useful to have a fresh set of eyes when doing this. But that's OK, because there are 8 co-authors so maybe one of them could help out.

- The key take home message is that the Zn isotope ratio of the boundary layer is heavier than the country rock and that this heavy signal can only come from impact debris that is isotopically heavy. However, I don't think this is well argued right now. There needs to be clear explanation that terrestrial weathering processes (e.g. clay formation) do not cause this type of fractionation. Current discussion probably touches on this but is poorly organized.

We agree with this comment. We have integrated the importance of the mixing trend and that we need to generate a high Zn isotope value and low Zn concentration end member. Focusing on these two important aspects of our data eliminates confusion and further strengthens our argument.

Here is the section reorganized:

Evidence for a distinct volatile Zn signature in boundary layer sediment

The salient observation of the current dataset is that the boundary layer sediments possess high $\delta^{66}\text{Zn}$ values that correlates with low Zn concentration. The Zn isotope composition of the K-Pg layers analyzed possess higher Zn isotope values than the surrounding sedimentary rocks (an increase of +0.7 per mil at the Caribbean site, +0.3 per mil at the Missouri and Montana sites +0.1 per mil at Mississippi and New Jersey sites (Table 1 and Figures 1 & 2)). Explanation of the origin of this signature requires a process that generates both characteristics. Multiple hypotheses could explain natural processes that generate higher Zn isotope values such as: melt-induced fractionation, inherited signatures from the target rocks where the meteor struck, meteoritic materials that settled at the boundary, secondary alteration/depositional processes, and residuum from volatilized material from the impact event that settled in the "fall-out" layers. However, a process producing the combination of higher Zn isotope values with low Zn concentration must occur to explain the trend observed here. The discussion below explores which of these processes might be capable of generating a high Zn isotope value and low Zn concentration end member, and/or could create a consistent Zn isotope signature over such large spatial scales,

Mechanisms relating to magmatic differentiation processes and associated with melting of target rocks at the impact site cannot generate the elevated Zn isotope values present in the boundary layer. Multiple studies of Earth's mantle melts, and their associated peridotite residues, reveal that Zn does not routinely fractionate greater than 0.2 per mil²⁴⁻²⁸ during magmatic differentiation at mantle potential temperatures of 1300 to 1500°C. Considering that the instantaneous melting temperature associated with impact matches or exceeds this range (e.g. >1300°C decelerative drag heating during terminal in-fall of ejecta²⁹), the production of impact melt alone is unlikely to produce the large Zn isotope fractionation measured herein (e.g. the Caribbean site samples).

Further evidence that magmatic processes did not cause the Zn isotope and concentration anomalies of the K-Pg reside in a comparison with the Permian-Triassic boundary (P-T). The P-T boundary possesses an abrupt increase of Zn concentration and a concomitant $\sim 0.5\%$ decrease in $\delta^{66}\text{Zn}$ occurring ~ 35 ky before the end-Permian mass extinction. The Zn isotopic and concentration changes have been interpreted to demonstrate rapid and massive input of isotopically light Zn from volcanic ashes, hydrothermal inputs, and/or extremely fast weathering of large igneous provinces (LIPs) into the oceans³⁰. In comparison to the P-T boundary, the elevated Zn isotope ratios and decrease of Zn concentration of the sedimentary rocks in K-Pg boundary sediments are clearly inconsistent with such inputs of isotopically light Zn from processes associated with extensive volcanism. Under this premise, a mass flux from volcanism and weathering associated with the Deccan Traps, the LIP which precedes the K-Pg boundary⁵, would only serve to dampen the positive $\delta^{66}\text{Zn}$ excursion at the boundary—not to generate or enhance it—especially in the marine settings.

Inheritance of the Zn isotope signature from target rocks does not explain the unique Zn signature in the K-Pg boundary layer. Extensive study of Zn isotope compositions in Earth materials over the last 20 years indicates that the vast majority of terrestrial geologic processes do not significantly fractionate Zn isotopes beyond $\sim 0.2\%$, with the exception of volatility-related processes. While certain mineral, gas, and biologically-derived materials, may possess Zn isotope compositions outside the range reported above (e.g. $\delta^{66}\text{Zn}$ values ranging from -1 to $+2\%$), bulk rock values containing such phases typically do not deviate from bulk silicate Earth (BSE) or continental crustal values, because these materials are often only present at trace levels at the macro/regional scale. This is most relevant for biogenic carbonates that can have slight enrichments in heavy Zn isotopes³¹, however these typically contain only ~ 1 ppm Zn³², while isotopically lighter (lower $\delta^{66}\text{Zn}$) continental margin and basin marine sediments (e.g. $\delta^{66}\text{Zn} \sim 0.12\%$) can contain 10s to 100s of ppm Zn³³.

Derivation of the Zn signature from meteorite impactor materials are equally improbable as a source for the elevated Zn isotope signatures within the K-Pg boundary layer because carbonaceous chondrites generally possess >100 ppm Zn, and typical Zn isotope values are less than $+0.5\%$ ³⁷ (with few outliers beyond this). The concentration data alone eliminate meteorites as a source, generally speaking, because the mixing trends observed at the boundary layer in that case would have a trend opposite to what is observed herein (a negative slope instead of the positive slope documented). Notable exceptions in the literature, with respect to high $\delta^{66}\text{Zn}$ values, are heated and/or thermally metamorphosed meteorites^{16,34-37}. However, such examples—e.g. impact-heated or thermally metamorphosed meteorites^{34,37} are extremely rare in the meteorite record, and these meteorites are also generally marked by low Zn concentrations (as they are themselves likely products of Zn volatilization). As a final point in this regard, Cr isotopes indicate a [CM2] carbonaceous chondrite-like material as the K-Pg bolide³⁸, thus excluding the latter possibility of a thermally metamorphosed enstatite chondrite as the high $\delta^{66}\text{Zn}$ source, since enstatite (and ordinary) chondrites have dramatically different Cr isotope signatures relative to carbonaceous chondrites³⁹. Given these observations, and the fact that impact ejecta is largely comprised of target material (with higher Zn concentrations), it is exceedingly unlikely (though of course not entirely impossible) that such materials are the high $\delta^{66}\text{Zn}$ source indicated herein.

Zn isotopic shifts related to secondary alteration processes are not significant enough to produce the Zn isotope excursions typically found at the boundary. Redox and other kinetic reactions associated with biogeochemical interactions in soils and shallow depositional environments produce relatively small Zn isotope fractionations (0.1 to 0.2 ‰^{33,40-42}), and measured Zn isotope fractionation among geochemical products and reactants rarely approach or exceed 0.3 ‰ in ore minerals and clays⁴³⁻⁴⁶. The stratigraphic changes in Zn isotopic signature of the K-Pg boundary sediments (Fig 1) also indicates minimal potential for terrestrial alteration as a cause for the higher values at the boundary. Significantly, adjacent sediment layers in the sections sampled show no element-isotope mixing systematics. This indicates that no significant leaching of heavy Zn isotopes from under- or overlying rock layers took place, arguing against this process as a cause of $\delta^{66}\text{Zn}$ enrichment within the boundary layer itself, and highlights the otherwise typical nature of the stratigraphic sequence with regards to Zn isotope composition. These observations agree with studies that have addressed secondary fractionation processes in sediments, which indicate that such processes do not fractionate Zn isotopes in a systematic way or to any large degree^{43,44,47}. As a final point here, it is unlikely that such local/regional scale processes would produce a consistent signal across the spatial scales and varied depositional environments considered herein.

All the above in tow, the most probable explanation is that the high $\delta^{66}\text{Zn}$ values coupled with low Zn concentration observed at the K-Pg boundary originate from partial volatilization of Zn during impact related processes (e.g. drag heating during ballistic outfall). Research on tektites—fused glasses from cooling of impact induced melts⁴⁸—along with complementary work on fused silicates from nuclear blast sites, and experimental Zn volatilization, show that Zn isotopes fractionate by as much as +0.5 to +2 ‰¹⁹⁻²¹ during Zn evaporation. For impact ejecta, the physical composition of projectiles is often that of the target material mixed with a smaller contribution from the impactor, and therefore the initial Zn isotope compositions of projectiles most likely mimics that of the target material. During impact events, ejecta such as proto-tektites, impact spherules, impact glass breccia and other materials display Zn loss from volatilization that augments the Zn isotope composition of the residual phase (i.e. increases $\delta^{66}\text{Zn}$ in the melt droplet), thus acting as a distinct mechanistic indicator of Zn loss via volatilization.

The takeaway from all the above is that, while other mechanisms cannot be entirely ruled out, it is probabilistically most logical that a high $\delta^{66}\text{Zn}$ value, low Zn concentration trend (such as that observed herein) acts as a multi-dimensional mechanistic tracer of Zn evaporation. Considering Zn volatilization as the operative process, the data can then be further scrutinized to elucidate details of material mixing during impact and sedimentation, and to compare the observed Zn volatilization trend (higher $\delta^{66}\text{Zn}$ with lower Zn concentration) with that for similar—though not identical—datasets in the literature, namely tektites.

There also needs to be better discussion of why this heavy signal could not come from meteorites. For reference, enstatite chondrites have $\delta^{66}\text{Zn}$ values up to +7permil (Moynier et al., 2011) so there should be thorough discussion of why impactors with such heavy Zn isotope ratios could not be responsible.

See above for explanation and evidence that we have included further discussion of such possibilities, and why are they unlikely. We are not using just the Zn isotope value. The concentration data are imperative in the explanation. We point to this again in the text.

Added text in response to this in “Evidence for a distinct volatile Zn signature in boundary layer sediment”:

The concentration data alone eliminate meteorites as a source, generally speaking, because the mixing trends seen the boundary layer would have an opposite trend (a negative slope instead of the positive slope documented). Notable exception in the literature, with respect to high $\delta^{66}\text{Zn}$ values, are heated and/or thermally metamorphosed meteorites (e.g. Mahan et al., 2018; Moynier et al. 2011, 2017, respectively). However, such examples are rare in the meteorite record, and are also generally marked by low Zn concentrations (as they are products of Zn volatilization). As a final point in this regard, Cr isotopes indicate a [CM2] carbonaceous chondrite-like material as the K-Pg bolide (REF# 7), thus excluding the possibility of a thermally metamorphosed enstatite chondrite as the high $\delta^{66}\text{Zn}$ source, as enstatite (and ordinary) chondrites have dramatically different Cr isotope signatures (e.g. Warren, 2011). Given these observations, and the fact that impact ejecta are largely comprised of target material (with higher Zn concentrations), it is exceedingly unlikely (though not impossible) that such materials are the high $\delta^{66}\text{Zn}$ source indicated herein....

...The takeaway from all the above is that, while other mechanisms cannot be entirely ruled out, their statistical likelihood is minimal. Moreover, a high $\delta^{66}\text{Zn}$, low Zn concentration trend (such as that observed herein) acts as a multi-dimensional mechanistic tracer, i.e. it is negative correlation between the two that acts as a robust process tracer of evaporation. Considering Zn volatilization as the operative process, the data can then be further scrutinized to elucidate details of material mixing during impact and sedimentation, and to compare the observed Zn volatilization trend (higher $\delta^{66}\text{Zn}$ with lower Zn concentration) with that for similar (though not identical) datasets in the literature, namely tektites.

- Figure 2 – I don’t understand the point of the figure. Comparing the sediments measured here with literature data from the continental crust doesn’t seem useful. You need a clear way to show that the boundary layer rocks have heavier $\delta^{66}\text{Zn}$ values than the adjacent sediment layers but this isn’t it.
- Check references are correct. Make sure that all tektite data is considered in addition to Moyniers 2009 paper.

While the reviewer seems to find the manuscript figures inadequate, no constructive feedback is given by the reviewer as to how the figures “should be” or could be improved. In any case, we have given considerable effort towards both revising and explaining the figures, both within these responses and in the revised manuscript.

We show in figure 1 that the boundary layers are different from the surrounding layers. Although Figure 1 has largely gone unchanged, this reviewer now sees its utility, thus it is also difficult to elucidate Reviewer #2’s criteria for a useful vs un-useful figure, however we are very glad that the reviewer now appreciates Figure 1.

In figure 2, we plot out the range of Zn isotope values known to this point for the material types being considered. This figure clearly shows that the K-Pg boundary layers are enriched in heavy Zn isotopes relative to continental crust (i.e. average target material), and lie on a continuum with gas condensates and tektites. We note that gas condensates can be excluded as a possible process/source, because these would have a correspondent increase in Zn concentration with higher $\delta^{66}\text{Zn}$. Thus, Figure 2 quite clearly displays that K-Pg boundary layers, when excluding condensates or exogenous/endogenous sources (argued elsewhere), are $\delta^{66}\text{Zn}$ enriched relative to continental crust, but depleted relative to tektites, and are therefore a mixture of materials akin to these two.

Finally, without justification for necessity or added utility (which Reviewer #2 does not give), we do not understand the request made by Reviewer #2 to include further tektite data, especially given the reviewer's posturing regarding figure clarity and utility in general. The tektite data of Wimpenny et al. (2019) do not add further insight, as these data were a supplement to the main focus of that work, which was to determine experimental Zn isotope fractionation during evaporative loss. These data cluster entirely within that of Moynier et al. (2009) in $\delta^{66}\text{Zn}$ vs [Zn] space (d66Zn range of approximately 1-2 per mil, all concentrations below ~ 30 ppm Zn), and thus their inclusion does not enhance Figure 2 (or Figure 3) in any way. Tektite (moldavite) data from Rodovská et al. (2017) show generally higher $\delta^{66}\text{Zn}$ values than that in Moynier et al. (2009) (approx.. 2-4 per mil, 110 ppm Zn or less). However, no defined explanation is given as the higher $\delta^{66}\text{Zn}$ values observed, limiting the utility of including and/or discussing these data. Furthermore, the addition of these data to Figure 2 or Figure 3 would require doubling the Y axis on both figures, which would only serve to reduce the visible comparison of all the other data.

To acknowledge the existence of these data, they have been appropriately cited in the figures and where relevant.

I think these issues are fairly straightforward to address although would involve some reorganization of the discussion which is non-trivial. My recommendation is that the author perform the changes under the guidance of the editor and that once complete the manuscript will be acceptable for publication.

L100- But to what degree is this heavier than rocks/sediments that have not experienced volatile loss? Can the volatile depletion signal be unambiguously identified? This is a key point regarding why you would want to use Zn isotopes for this study.

To this point, we have already shown that the boundary layer is enriched in heavy Zn isotopes. We possess no data to constrain exactly how much volatilization occurred, or how much partially volatilized material is in the layer. Despite this, the high $\delta^{66}\text{Zn}$ / low Zn concentration trend is incredibly clear and provides the most unambiguous signal to date for a volatilization event at the K-Pg boundary, and because of the unknown variables mentioned above, the data are qualitative and therefore we do not attempt to quantify the extent of volatilization relative to other rocks/sediment (we feel this would be scientifically faulty).

L145 - Figure 2 doesn't show this. It shows that the sediments and spherules contain heavier Zn isotope ratios than rocks from the continental crust (on average).

In the first revision of our manuscript, we added figure 1 as a means to further clarify this statement. In this way it is comparative to all known Zn isotope value from the literature for material types being considered, and the surrounding rock layers presented in this document.

We do not understand the necessity to compare to ALL tektite data in existence. Reviewer #2 seems very particular with regards to figures and values figure clarity. The tektite data from Moynier et al. (2009) is quite representative of that seen in the literature (e.g. Wimpenny et al., 2019), while the data in Rodovská et al. (2017) contain higher $\delta^{66}\text{Zn}$ values that are so far unexplained. Including the former would add no additional value to the figures and only obscure the trend detailed by Moynier et al. (2009); including the latter would require expanding the Y axis roughly two-fold, thus severely obscuring the reader's ability to visibly track the heavy isotope enrichment as Zn concentration decreases. In the caption to Figure 3, we have now added both of these references—which combined with Moynier et al. (2009) comprise published tektite data for Zn isotopes—and have also added justification for not including them in the figure. In this way, we are duly acknowledging the existence of these published works, while also displaying the data in such a way that it can be visually interpreted (the main point of a figure).

L147 - How do you know the composition of the bolide? Meteorites have a wide range of Zn isotope ratios, some of which are very heavy (e.g. Moyniers work on enstatite chondrites).

Agreed, we added language to acknowledge this in the comment above and specifically reference the +7 per mil outlier that exists in this dataset. We also acknowledge that heated meteorites can also have heavy Zn isotope enrichments, and in both cases, this is considered to be due to volatile loss.

That said, the argument made here by the reviewer is indefensible given current published datasets. To start, the enstatite chondrite example given only considers Zn isotopes in a contextual vacuum, as Cr isotopes from the K-Pg boundary are similar to carbonaceous chondrites, and thus dramatically different to that in enstatite chondrites. All we can do is take all the available data and information, and then discuss the most PROBABLE cause or causes. To discuss unlikely scenarios to any significant degree creates a false equivalency of likelihood and/or importance, and thus would be scientifically misguided.

L152 - These points aren't on Figure 2.

Deleted figure 2 reference here

L155 - This is not correct. Enstatite chondrites have Zn isotope ratios up to +7 permil heavier than the bulk silicate earth (Moynier et al., 2011).

As stated above, we reference such outliers, and the same for heated meteorites. We now also explicitly point out that while these can't be ruled out, as they are statistically unlikely (as "deus ex machina" explanations) and therefore not discussed in detail. We note also (here and in the revised manuscript)

that for enstatite chondrites, the Cr isotope data for the K-Pg boundary quite definitively exclude this possibly.

L159 - Only one sample is relatively heavy. The rest have $\delta^{66}\text{Zn}$ values that are found in crustal rocks that have not experienced volatilization. I think it is more important to stress the isotopic difference between the boundary layer and the country rocks. Can you resolve an isotopic difference in each profile? What is that difference?

There is a clear mixing relationship here with volatilized Zn, we establish this more clearly in the argument. The authors would also like to—again—point out that there are exceedingly few examples in the literature where Zn isotope values are above +0.6, one in a black shale, a few thermally metamorphosed enstatite chondrite meteorites, and a couple in heated meteorites (which Reviewer #2 does not seem to be aware of). Both of these Zn reservoirs have higher Zn concentrations and cannot resolve the concentration data provided here. As stated above, we point to this now.

L163 - Didn't you discount these processes in the previous paragraph?

See above in the reorganized section. We have also now noted in the revised manuscript where we reiterate previous information for further perspective and as context.

L174 - But isn't the point here that heating (and melting) was accompanied by degassing of isotopically light Zn? Are you talking about magmatic differentiation processes (i.e. mineral-melt partitioning)?

Changed wording to magmatic differentiation processes on line 167

Mechanisms relating to the magmatic differentiation processes associated with melting

L182 – What about basic silicate weathering processes like clay formation?

Added this language and updated references

Measured Zn isotope fractionation among geochemical products and reactants rarely approach or exceed 0.3 ‰ in ore minerals and clays^{27,40-42}.

L197 – Is this process even likely to happen? Is there any evidence that leaching of isotopically heavy Zn from sediment layers can impart a heavy signature to neighbouring rocks? It seems overly complicated. Wouldn't a simpler scenario be that a period of intense weathering leached away isotopically light Zn from the boundary layer?

The authors find it highly unlikely that weathering solutions strip lighter Zn similarly in all different sites that have different depositional environments, local conditions (e.g. pH, T), etc.? Furthermore, no known depositional process produces a +0.7 per mil difference during weathering, which would need to be the case to explain the Caribbean site? Lastly, so-called “extreme” weathering tends to preferentially

strip out heavy Zn isotopes, if at all, contrary to the mechanism proposed by Reviewer #2 (Little et al., 2019).

The authors sincerely appreciate that there are other mechanisms that could be the source/cause of the heavy Zn isotope enrichment in the boundary layer, and we acknowledge these ad nauseum in the manuscript. However, these mechanisms suggested by Reviewer #2, and others referenced within the manuscript, are highly scientifically and statistically unlikely, and therefore are not discussed in greater detail.

L203 – A problem here is that there hasn't been enough effort to unambiguously show that the boundary layers have higher Zn isotope ratios than the country rock. Figure 2 doesn't show this (despite what is claimed on L145). And Fig. 1 is too small to see easily. Is the isotopic difference between boundary layer and country rock always statistically significant? Also, there is no clear argument provided that material from the impactor itself couldn't have contributed isotopically heavy Zn to the boundary layer. These should be key parts of the discussion. The rest of the section (L144-203) is quite muddled. You start off by stating that target rocks and meteorites don't contain isotopically heavy Zn that could explain the single heaviest $\delta^{66}\text{Zn}$ value (L150-159) and then go on to repeat some of this discussion later (L176-192). This section should be consolidated and discussion of meteorites expanded to include samples such as enstatite chondrites.

As mentioned above, the impactor itself does not make up much of the ejecta material (10% or less), and that is consistent with what's observed in the current work. Furthermore, if this were the cause, it would mean that the sediments would have to accumulate differing amounts of this meteoritic component that corresponded very well to the Zn isotope concentrations; this is incredibly unlikely given the ballistic trajectory these materials were on and the heating/volatilization incurred.

Also as mentioned above, we note where we have re-iterated information for further perspective and for context within the given section, and we believe this is merited.

Finally, and again previously noted in argument against enstatite chondrites as a possible source, we strongly disagree with expanding discussion of meteorites, and especially for enstatite chondrites. This is because the K-Pg boundary has a Cr isotope anomaly that clearly shows mixing between a terrestrial component ($\epsilon^{54}\text{Cr} \sim 0$) and an impactor with higher $\epsilon^{54}\text{Cr}$. Since enstatite chondrites (and ordinary chondrites) have negative $\epsilon^{54}\text{Cr}$ signatures (and carbonaceous chondrites have positive $\epsilon^{54}\text{Cr}$ signatures), this excludes enstatite chondrites as a viable source and promotes a carbonaceous chondrite as a source. While heated meteorites of the carbonaceous chondrite variety do exist (see Mahan et al., 2018), these are very rare and also likely are derived from outer shell material of the parent body, making them both unlikely and probably insignificant with respect to being a high $\delta^{66}\text{Zn}$ source.

L203 – I do agree that the heavy $\delta^{66}\text{Zn}$ values at the boundary layers are consistent with the addition of volatile depleted material. I just don't think the current discussion has been optimized to show this.

We have endeavored to use these previous comments to meaningfully improve the manuscript and refine our arguments, and truly hope that these revisions and responses content Reviewer #2.

L208 – The key observation is relatively simple; the increase in $\delta^{66}\text{Zn}$ correlates with a decrease in Zn concentration. Is this what is seen in the K-Pg boundary sediments?

Yes, this is exactly what is seen and what has been stated at multiple points in the manuscript, original and revised. We have endeavored to make this even more explicit in the revised manuscript. To emphasize this more we changed the whole section to make this clearer.

L220 – Why would fast weathering impart a light signature to the oceans? Does this mean that weathering residue is isotopically heavy? Why is volcanic ash isotopically light? What process causes this? What are the implications of this for interpretation of K-Pg boundary sediments?

While the authors appreciate the Reviewers questions, we do not see where this string of questioning is going. The answers to these questions lie within the reference given (Liu et al., 2017), and this is common convention when acknowledging other potential processes that may occur but are either unlikely and/or not viable within the context. The implications are exactly what we say they are in the original and revised manuscript, that is, the Zn isotope/concentration relationship related to large-scale volcanism (and the ash, weathering that comes with this) yield an opposite trend to what is seen in the K-Pg sediment. Beyond this, it is unclear what more the reviewer is asking for, as all these questions have already been answered in the manuscript and references therein.

L246 – The reference for the isotopic composition of Zn in tektites is incorrect. Fig. 3 references Moyniers 2009 paper but there are at least 3 studies that I know of that contain Zn isotope data from tektites, and possibly more exist. They should all be referenced in this study. I said this in my previous review.

Adding the other two datasets being references (Rodovská et al., 2017; Wimpenny et al., 2019) do not significantly enhance anything. Adding data and references for their own sake is not a reason. We have robustly defended our position in responses above. However, in response to Reviewer #2's comments, we have added brief discussion of these datasets within the revised manuscript in acknowledgement of their existence.

L266 – This is because different tektites have experienced different extents of Zn loss.

Added

because tektites experienced different amounts of Zn loss^{6,13,45,46}

L290-299 – I don't really understand the point being made here. It seems to repeat the point made on L282 about mixing of ejecta with sediment layers. I'd just delete these lines and skip to the final sentence on L299 about increasing $\delta^{66}\text{Zn}$ values corresponding to decreasing Zn concentrations.

Deleted these lines

L304-316 and L317-330 – These paragraphs repeat each other and also repeat discussion on L245-254 about contamination of the impact signal by sediment input. These should be consolidated. I honestly don't understand - did nobody read this?!

Deleted these lines

L335 – But if carbonates don't contain much Zn how can they be the source of the isotopically heavy Zn in the boundary layer? Couldn't you rule them out based on this?

The boundary layer samples contain carbonate, but also impact spherules and other sedimentary components. In the original and revised manuscripts we acknowledge that it is not possible to differentiate between the target carbonates and gneissic basement materials. We go on to point out that the carbonates have much lower Zn concentrations (~1 ppm), while margin and basin sediments, and continental crust, can have 10s to 100s, and thus these likely constitute the overall Zn budget of the ejecta (not carbonate platform)

L341-358 – The suppression of alpha was nicely explained on L265-289. Its unnecessary to repeat that here. Again, these sections should be consolidated.

We purposefully reiterate some of the discussion of element volatilization, and suppressed isotopic fractionation under terrestrial conditions in the section "Implications of the mass evaporation event and Zn mass balance" as context, providing a means to expand the discussion to include other elements and the larger discussion about the volatile element budget of Earth. However, we agree with Reviewer #2 that parts of this discussion could be appreciably condensed. In the revised manuscript, we have removed redundant discussion of Zn volatilization and isotope fractionation so that this section is more concise and is now more honed in on the discussion of Earth's volatile element budget.

L363-369 – Break this sentence up.

Reworded

L372 – This is a long winded way of saying that impacts don't contribute enough Zn mass to perturb the bulk silicate earth Zn composition. Are there estimates of total extraterrestrial material to have hit the Earth (vs total silicate mass)?

This discourse was in response to another reviewer who expressed a strong desire for further text on this, and as such this paragraph was added. We believe that one paragraph is not a lot of text at all considering the contextual breadth of information being covered.

That said, this paragraph is saying that it is very unlikely that volatile loss (from the Earth) due to impact has resulted in appreciable loss of Zn (as it's light isotopes) to space (as argued in the revised manuscript). It follows then that the isotopically light Zn stayed on Earth and must be somewhere, thus leading into the discourse around the isotopically light sample at the top of the Caribbean ODP core as a possible reservoir for this isotopically light Zn.

With the above in tow, an estimate of extraterrestrial material added to Earth over its lifespan is not actually germane to the current work. But for fun, and even though we were unable to find an estimate for the total extraterrestrial mass added to other over the past ~4.5 billion years, we can take NASA's calculated average of ~40,000 tonnes being added per year (40,000,000 kg), multiply this by

4,543,000,000 to yield 1.82×10^{17} kg, which must be a minimum given that impacts were more frequent earlier in Earth's history. Taking a BSE mass of $\sim 4 \times 10^{24}$ (assuming Earth mass of 5.97×10^{24} and core of 32.5 % of Earth's mass), yields $\sim 5 \times 10^{-8}$, or much less than 1 billionth of Earth's silicate mass.

L375 – What is the time lag between the light sample at the top of the boundary and the heavy sample at the base of the boundary, which is presumably from when the impact happened? I assume that amount of sedimentation represents a decent amount of geological time (hundreds/thousands of years?) Would you expect fine particles with isotopically light Zn isotope ratios to float around the atmosphere for such a long time? This all seems like a highly speculative hypothesis. Could a more mundane explanation such as a shift in weathering/climate/ocean conditions following the impact explain a drop in the Zn isotope ratio?

In short, the proposed mechanism is unlikely as such processes in general do not cause shifts in Zn isotope compositions, and the shifts they do cause can negatively interfere (meaning not all would create a light isotope excursion, e.g. weathering could be light or heavy). Furthermore, because of the differences in depositional environments, and the disruption in global ocean circulation and atmospheric patterns associated with the K-Pg, it is outside the scope of the current work to attempt to determine the time lag between the top and bottom intervals. Further to this, a global shift such as that suggested would suggest that the same isotope excursion should be seen at all sites, more-or-less, and this is not the case. Whereas ash fallout is inherently heterogeneous, and may help explain why it is not seen everywhere.

Lastly, we explicitly acknowledge in the original and revised manuscript that this is a very limited interpretation.

Reviewer #3 (Remarks to the Author):

The revision addresses all of the points raised in review and I have no further comments, and recommend acceptance of this manuscript.

Best regards,
Christian Koeberl
(Univ. Vienna, Austria)

Reviewers' comments:

Reviewer #1 (Remarks to the Author)

As a reviewer of the original submission, I feel that the majority of my concerns/suggestions have been addressed satisfactorily. For my taste, the modelling of the revised version is a bit too strongly focused on tektites. I think the general readers of the journal would benefit from a reminder that tektites are mainly splash curtain ejecta but that their isotope systematics and volatilisation history may not be perfect analogues for the bulk of the melt sheet, which contains much more overall mass. As a partly submerged impact structure Chicxulub almost certainly experienced explosive ejection of melt sheet material upon re-entry of seawater into the crater and I'm therefore not sure that tektites necessarily represent the best analogue for the extent of isotope fractionation that might have been experienced by the ejecta material found in the analysed sedimentary rocks. I'm not asking for a different model, simply for a statement that better explains why tektites per se may not be the best proxies for expected isotope ratios in the isotopically heavy end-member.

Reviewer #2 (Remarks to the Author)

This is a revised version of a manuscript that I previously reviewed for Nature Communications. Its an interesting subject and I'm sure would be of interest to readers of this journal. The observation that boundary layers from the K-Pg boundary are depleted in Zn and have a heavier Zn isotope ratio than the country rock is consistent with input of volatile depleted material from an impact event. So I do agree with the authors about the main thrust of the manuscript. However, there are still problems with it. The main problems being the following:

- Poorly organized. I made several comments saying that paragraphs could be consolidated due to repetition. The best example being paragraphs on L304 and 316. Honestly, this paper could be $\sim 1/4$ shorter and tightening it up would greatly improve the readability. Sometimes its useful to have a fresh set of eyes when doing this. But that's OK, because there are 8 co-authors so maybe one of them could help out.
- The key take home message is that the Zn isotope ratio of the boundary layer is heavier than the country rock and that this heavy signal can only come from impact debris that is isotopically heavy. However, I don't think this is well argued right now. There needs to be clear explanation that terrestrial weathering processes (e.g. clay formation) do not cause this type of fractionation. Current discussion probably touches on this but is poorly organized. There also needs to be better discussion of why this heavy signal could not come from meteorites. For reference, enstatite chondrites have $\delta^{66}\text{Zn}$ values up to +7permil (Moynier et al., 2011) so there should be thorough discussion of why impactors with such heavy Zn isotope ratios could not be responsible.
- Figure 2 – I don't understand the point of the figure. Comparing the sediments measured here with literature data from the continental crust doesn't seem useful. You need a clear way to show that the boundary layer rocks have heavier $\delta^{66}\text{Zn}$ values than the adjacent sediment layers but this isn't it.
- Check references are correct. Make sure that all tektite data is considered in addition to Moyniers 2009 paper.

I think these issues are fairly straightforward to address although would involve some reorganization of the discussion which is non-trivial. My recommendation is that the author perform the changes under the guidance of the editor and that once complete the manuscript will be acceptable for publication.

L100- But to what degree is this heavier than rocks/sediments that have not experienced volatile loss? Can the volatile depletion signal be unambiguously identified? This is a key point regarding why you would want to use Zn isotopes for this study.

L145 - Figure 2 doesn't show this. It shows that the sediments and spherules contain heavier Zn isotope ratios than rocks from the continental crust (on average).

L147 - How do you know the composition of the bolide? Meteorites have a wide range of Zn isotope ratios, some of which are very heavy (e.g. Moyniers work on enstatite chondrites).

L152 - These points aren't on Figure 2.

L155 - This is not correct. Enstatite chondrites have Zn isotope ratios up to +7 permil heavier than the bulk silicate earth (Moynier et al., 2011).

L159 - Only one sample is relatively heavy. The rest have $\delta^{66}\text{Zn}$ values that are found in crustal rocks that have not experienced volatilization. I think it is more important to stress the isotopic difference between the boundary layer and the country rocks. Can you resolve an isotopic difference in each profile? What is that difference?

L163 - Didn't you discount these processes in the previous paragraph?

L174 - But isn't the point here that heating (and melting) was accompanied by degassing of isotopically light Zn? Are you talking about magmatic differentiation processes (i.e. mineral-melt partitioning)?

L182 - What about basic silicate weathering processes like clay formation?

L197 - Is this process even likely to happen? Is there any evidence that leaching of isotopically heavy Zn from sediment layers can impart a heavy signature to neighbouring rocks? It seems overly complicated. Wouldn't a simpler scenario be that a period of intense weathering leached away isotopically light Zn from the boundary layer?

L203 - A problem here is that there hasn't been enough effort to unambiguously show that the boundary layers have higher Zn isotope ratios than the country rock. Figure 2 doesn't show this (despite what is claimed on L145). And Fig. 1 is too small to see easily. Is the isotopic difference between boundary layer and country rock always statistically significant? Also, there is no clear argument provided that material from the impactor itself couldn't have contributed isotopically heavy Zn to the boundary layer. These should be key parts of the discussion. The rest of the section (L144-203) is quite muddled. You start off by stating that target rocks and meteorites don't contain isotopically heavy Zn that could explain the single heaviest $\delta^{66}\text{Zn}$ value (L150-159) and then go on to repeat some of this discussion later (L176-192). This section should be consolidated and discussion of meteorites expanded to include samples such as enstatite chondrites.

L203 - I do agree that the heavy $\delta^{66}\text{Zn}$ values at the boundary layers are consistent with the addition of volatile depleted material. I just don't think the current discussion has been optimized to show this.

L208 - The key observation is relatively simple; the increase in $\delta^{66}\text{Zn}$ correlates with a decrease in Zn concentration. Is this what is seen in the K-Pg boundary sediments?

L220 - Why would fast weathering impart a light signature to the oceans? Does this mean that weathering residue is isotopically heavy? Why is volcanic ash isotopically light? What process causes this? What are the implications of this for interpretation of K-Pg boundary sediments?

L246 - The reference for the isotopic composition of Zn in tektites is incorrect. Fig. 3 references Moyniers 2009 paper but there are at least 3 studies that I know of that contain Zn isotope data from tektites, and possibly more exist. They should all be referenced in this study. I said this in my previous review.

L266 - This is because different tektites have experienced different extents of Zn loss.

L290-299 - I don't really understand the point being made here. It seems to repeat the point made on L282 about mixing of ejecta with sediment layers. I'd just delete these lines and skip to the final sentence on L299 about increasing $\delta^{66}\text{Zn}$ values corresponding to decreasing Zn concentrations.

L304-316 and L317-330 - These paragraphs repeat each other and also repeat discussion on L245-254 about contamination of the impact signal by sediment input. These should be consolidated. I honestly don't understand - did nobody read this?!

L335 - But if carbonates don't contain much Zn how can they be the source of the isotopically heavy Zn in the boundary layer? Couldn't you rule them out based on this?

L341-358 - The suppression of alpha was nicely explained on L265-289. It's unnecessary to repeat that here. Again, these sections should be consolidated.

L363-369 - Break this sentence up.

L372 - This is a long winded way of saying that impacts don't contribute enough Zn mass to perturb the bulk silicate earth Zn composition. Are there estimates of total extraterrestrial material to have hit the Earth (vs total silicate mass)?

L375 - What is the time lag between the light sample at the top of the boundary and the heavy sample at the base of the boundary, which is presumably from when the impact happened? I assume that amount of sedimentation represents a decent amount of geological time (hundreds/thousands of years?) Would you expect fine particles with isotopically light Zn isotope ratios to float around the atmosphere for such a long time? This all seems like a highly speculative hypothesis. Could a more mundane explanation such as a shift in weathering/climate/ocean conditions following the impact explain a drop in the Zn isotope ratio?

Reviewer #3 (Remarks to the Author)

The revision addresses all of the points raised in review and I have no further comments, and recommend acceptance of this manuscript.

Best regards,

Christian Koeberl

(Univ. Vienna, Austria)